# Implementation and benchmarking of a novel analytical framework to clinically evaluate tumor-specific fluorescent tracers

Marjory Koller[1], Si-Qi Qiu[1,2], Matthijs D. Linssen[3,4], Liesbeth Jansen[1], Wendy Kelder[5], Jakob de Vries[1], Inge Kruithof[6], Guo-Jun Zhang[7], Dominic J. Robinson[8], Wouter B. Nagengast[3], Annelies Jorritsma-Smit[4], Bert van der Vegt[9] & Gooitzen M. van Dam[1,10,11]

During the last decade, the emerging field of molecular fluorescence imaging has led to the development of tumor-specific fluorescent tracers and an increase in early-phase clinical trials without having consensus on a standard methodology for evaluating an optical tracer. By combining multiple complementary state-of-the-art clinical optical imaging techniques, we propose a novel analytical framework for the clinical translation and evaluation of tumor-targeted fluorescent tracers for molecular fluorescence imaging which can be used for a range of tumor types and with different optical tracers. Here we report the implementation of this analytical framework and demonstrate the tumor-specific targeting of escalating doses of the near-infrared fluorescent tracer bevacizumab-800CW on a macroscopic and microscopic level. We subsequently demonstrate an 88% increase in the intraoperative detection rate of tumor-involved margins in primary breast cancer patients, indicating the clinical feasibility and support of future studies to evaluate the definitive clinical impact of fluorescence-guided surgery.

[1] Department of Surgery, University Medical Center Groningen, University of Groningen, Groningen 9700 RB, The Netherlands. [2] The Breast Center, Cancer Hospital of Shantou University Medical College, Shantou 515000 Guangdong, China. [3] Department of Gastroenterology and Hepatology, University Medical Center Groningen, University of Groningen, Groningen 9700 RB, The Netherlands. [4] Department of Clinical Pharmacy and Pharmacology, University Medical Center Groningen, University of Groningen, Groningen 9700 RB, The Netherlands. [5] Department of Surgery, Martini Hospital, Groningen 9700 RM, The Netherlands. [6] Department of Pathology, Martini Hospital, Groningen 9700 RM, The Netherlands. [7] Changjiang Scholar's Laboratory of Shantou University Medical College, 515000 Shantou, Guangdong, China. [8] Erasmus Medical Center Rotterdam, Rotterdam 3015 GD, The Netherlands. [9] Department of Pathology, University Medical Center Groningen, University of Groningen, Groningen 9700 RB, The Netherlands. [10] Department of Nuclear Medicine and Molecular Imaging, University Medical Center Groningen, University of Groningen, Groningen 9700 RB, The Netherlands. [11] Department of Intensive Care, University Medical Center Groningen, University of Groningen, Groningen 9700 RB, The Netherlands. These authors contributed equally: Marjory Koller, Si-Qi Qiu. Correspondence and requests for materials should be addressed to G.Dam. (email: g.m.van.dam@umcg.nl)

Molecular fluorescence imaging enables visualization of tumor-specific, upregulated proteins and biological processes involved in oncogenesis and allows real-time imaging of tumor tissue with a high resolution for various clinical applications, such as image-guided surgery, endoscopy, and pathology. During the past decade, several tumor-specific fluorescent tracers have been developed and validated in animal models, leading more recently to a substantial increase in early-phase clinical trials evaluating molecular fluorescence imaging[1,2]. Despite the increasing activity in the field, several critical factors to ensure translation of optical tracers to clinical applications remain insufficiently established. No widely accepted analytical framework or standard evaluation methodology serves as a gold standard for determining the efficacy of a fluorescent tracer in clinical applications.

The majority of these early-phase clinical studies have been executed in image-guided surgery applications. In oncological surgery, it is important to remove the tumor completely without any residual disease, since incomplete resections are inevitably associated with higher rates of re-operations, increased rates of recurrent disease and lower overall survival[3]. Intraoperatively, surgeons are mainly dependent on visual inspection and palpation alone to distinguish cancer tissue from benign tissue, a method with unknown accuracy. The available intraoperative techniques for margin assessment have not yet been adopted universally. Frozen section analysis and imaging techniques like specimen radiography are time consuming and lack diagnostic accuracy[4]. Anatomical imaging modalities like CT and MRI have been adapted for use in the operating theatre, but cannot be used in real-time and have limited tumor specificity. Consequently, there is an unmet need for real-time tumor-specific imaging that is compatible with the workflow in the operating theatre. This might be provided by using sensitive optical imaging techniques combined with tumor-specific fluorescent tracers; this approach is currently under investigation in early-phase clinical trials[1].

As there is no consensus on a standard evaluation methodology for fluorescence imaging, we implemented a novel analytical framework for data collection, and fluorescence image analyses based on our experience in molecular fluorescence imaging in surgery and endoscopy[5–8]. In the present study we implement this novel analytical framework and confirm the tumor-specific targeting of the near-infrared (NIR) fluorescent tracer bevacizumab-800CW in escalating doses on a macroscopic and a microscopic level. We subsequently observed an 88% increase in the intraoperative detection of tumor-involved margins, thus indicating clinical feasibility in support of future studies to evaluate the definitive clinical impact of fluorescence-guided surgery in primary breast cancer patients.

## Results

**Summary of study design and patient demographics.** The study was designed as a clinical dose escalation trial investigating four doses of bevacizumab-800CW (4.5 mg, 10 mg, 25 mg, and 50 mg) in patients with invasive T1–T2 primary breast cancer scheduled for breast cancer surgery. Bevacizumab-800CW was injected intravenously three days prior to surgery (Fig. 1a). A step-up approach was used in which three patients per dose group were included, followed by expansion of the two best-performing dose groups to a total of ten patients each (Supplementary Fig. 1). Twenty-six patients with invasive primary breast cancer were enrolled between 12 October 2015 and 2 February 2017. Three patients received 4.5 mg, ten patients 10 mg, ten patients 25 mg, and three patients 50 mg of bevacizumab-800CW. No serious adverse events, allergic or anaphylactic reactions, were reported in this trial. Two adverse events were reported, one patient from the 4.5 mg group experienced nausea till 30 min after tracer injection, another patient from the 25 mg group had hot flushes after tracer administration that recovered spontaneously. None of the patients felt any burden of the infusion three days prior to surgery.

In 19 patients, histopathological analyses showed an invasive carcinoma of no special type (NST); in five patients a lobular carcinoma, in one patient a mucinous carcinoma and in one patient a papillary carcinoma. In four patients, there was a tumor-involved surgical margin of the invasive primary tumor; in four other patients the surgical margin of unexpected additional carcinoma in situ was positive adjacent to a completely removed primary tumor. This resulted in a total positive-margin rate of 30% according to the most recent SSO-ASTRO guidelines[9] (Table 1).

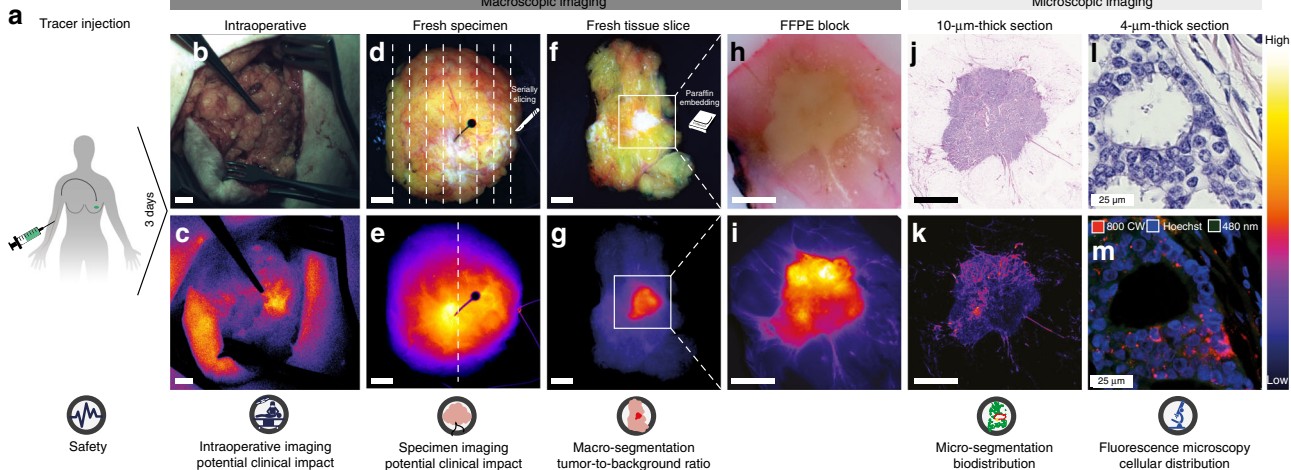

**Fig. 1** The clinical analytical framework enabling correlation of intraoperative fluorescence signals with histopathology, from macroscopic to microscopic levels. **a** Intravenous administration of bevacizumab-800CW three days prior to surgery. **b, c** Color image and corresponding fluorescence image obtained in vivo during surgery to determine potential clinical value. **d, e** Imaging of the fresh surgical specimen, followed by serially slicing. **f, g** Imaging of the fresh tissue slices to determine tumor-to-background ratio based on macro-segmentation, followed by paraffin embedding. **h, i** Imaging of formalin-fixed paraffin-embedded (FFPE) blocks to determine heterogeneity of tracer uptake within a tumor. **j, k** Imaging of 10-μm-thick tissue sections for micro-segmentation to reveal microscopic biodistribution and correlation with fluorescence signals from the macroscopic to microscopic level. **l, m** Fluorescence microscopy to determine tracer distribution on a cellular level. Scale bars represent 1 cm, in **l, m** the scale bar represents 25 μm

**Table 1 Demographics of study patients**

| | 4.5 mg $n=3$ | 10 mg $n=10$ | 25 mg $n=10$ | 50 mg $n=3$ |
|---|---|---|---|---|
| Patient characteristics | | | | |
| Age (years/range) | 72 (68–77) | 61 (50–69) | 57 (49–69) | 63 (53–70) |
| Clinicopathological parameters (number) | | | | |
| Tumor type invasive primary tumor | | | | |
| Invasive carcinoma of no specific type | 2 | 9 | 6 | 2 |
| Lobular caricnoma | 0 | 1 | 4 | 0 |
| Mucinous carcinoma | 0 | 0 | 0 | 1 |
| Papillary carcinoma | 1 | 0 | 0 | 0 |
| Tumor size (cm/range) | 1.5 (1.4–1.7) | 1.3 (0.5–2.4) | 1.8 (0.7–3.2) | 0.9 (0.8–1.1) |
| Tumor grade (modified Bloom-Richardson) | | | | |
| Grade I | 0 | 4 | 0 | 0 |
| Grade II | 3 | 4 | 7 | 0 |
| Grade III | 0 | 2 | 3 | 2 |
| n/a | — | — | — | 1 |
| Estrogen receptor positive (>10%) | 3 | 9 | 8 | 3 |
| Progesterone receptor positive (>10%) | 2 | 8 | 7 | 2 |
| HER2 receptor positive | | | | |
| (IHC 2 + or 3 + with positive FISH) | 1 | 1 | 1 | 1 |
| Carcinoma in situ present | 1 | 6 | 9 | 3 |
| Safety data (number) | | | | |
| Adverse events | 1 | 0 | 1 | 0 |
| Serious adverse events | 0 | 0 | 0 | 0 |
| Surgical resection margin status (number)[a] | | | | |
| Primary tumor | | | | |
| Free | 3 | 7 | 9 | 3 |
| Not free | 0 | 3 | 1 | 0 |
| Additional Carcinoma in situ component | | | | |
| Free | 1 | 5 | 7 | 2 |
| Not free | 0 | 1 | 2 | 1 |

[a] Denotes according to ASTRO guidelines
*HER2* human epidermal growth factor receptor 2, *IHC* immunohistochemistry, *FISH* fluorescence in situ hybridization

**The analytical framework**. Breast-conserving surgery consists of intraoperative assessment of the margins by the surgeon using visual inspection and palpation and evaluation of the excised specimen by standard histopathology. Therefore, the data collection procedure within the analytical framework to determine the tumor-specific targeting of bevacizumab-800CW should fit the standard workflow and needs to provide qualitative and quantitative data on fluorescence related to the standard of care. As such, the data collection procedure within the analytical framework consisted of: (i) qualitative in vivo intraoperative macroscopic imaging to determine the potential clinical value of fluorescence-guided surgery (Fig. 1b, c), (ii) qualitative ex vivo imaging of the fresh whole surgical specimens (Fig. 1d, e), (iii) quantitative ex vivo imaging of the fresh tissue slices of the fresh specimens to determine the tracer distribution on a macroscopic level (Fig. 1f, g), (iv) quantitative multi-diameter single-fiber reflectance/single-fiber fluorescence (MDSFR/SFF) spectroscopy of the fresh tissue slices to determine the intrinsic fluorescence intensities, (v) quantitative fluorescence flatbed scanning of formalin-fixed paraffin-embedded (FFPE) blocks and 10-μm-thick sections to determine the tracer distribution on a microscopic level (Fig. 1h–k), and (vi) fluorescence microscopy (Fig. 1l, m).

**Quantitative macro-segmentation of the fresh tissue slices**. In all patients, qualitative assessment of fluorescence signals showed higher fluorescence signal intensities in tumor tissue compared to normal surrounding tissue at all ex vivo imaging modalities, including the fresh tissue slices, FFPE blocks and 10-μm-thick sections. Representative images per dose group and per imaging

modality are shown in Fig. 2a–x. A complete overview per patient is shown in Supplementary Fig. 2. Sodium dodecyl sulfate polyacrylamidegel electrophoresis (SDS-PAGE) of tumor lysates demonstrated that the complete compound bevacizumab-800CW was intact and present in the human primary breast tumor, as confirmed by comparing the height of the band of the tumor lysates with the lane containing diluted bevacizumab-800CW (Supplementary Fig. 3).

We used fluorescence images of all fresh tissue slices obtained by a light-tight macroscopic imaging device for quantitative macro-segmentation analyses to determine the tumor-specific targeting of bevacizumab-800CW on a macroscopic level by calculating the tumor-to-background ratio (TBR). Freshly excised tissue represents the in vivo situation the best for the calculation of the TBR, as it has not yet been processed with formalin or embedded in paraffin and the conditions of imaging are the most optimally standardized; i.e., the tumors within the slices are all on the surface without overlaying tissue, the distance from stage to camera is equal in all patients, and no ambient light is influencing the fluorescent signals. In the fluorescence images of all fresh tissue slices that contained tumor tissue after confirmation with histology, regions-of-interests of tumor tissue, as well as background tissue were manually segmented. The mean fluorescence intensity (MFI) was measured per region-of-interest (ROI) and averaged per tissue type, resulting in an MFI of tumor tissue and an MFI of background tissue per patient. TBR was calculated as the ratio of MFI of tumor compared to the MFI of normal tissue. Macro-segmentation analyses were performed on a total of 69 fresh tissue slices from 23 patients. In three patients, the light-

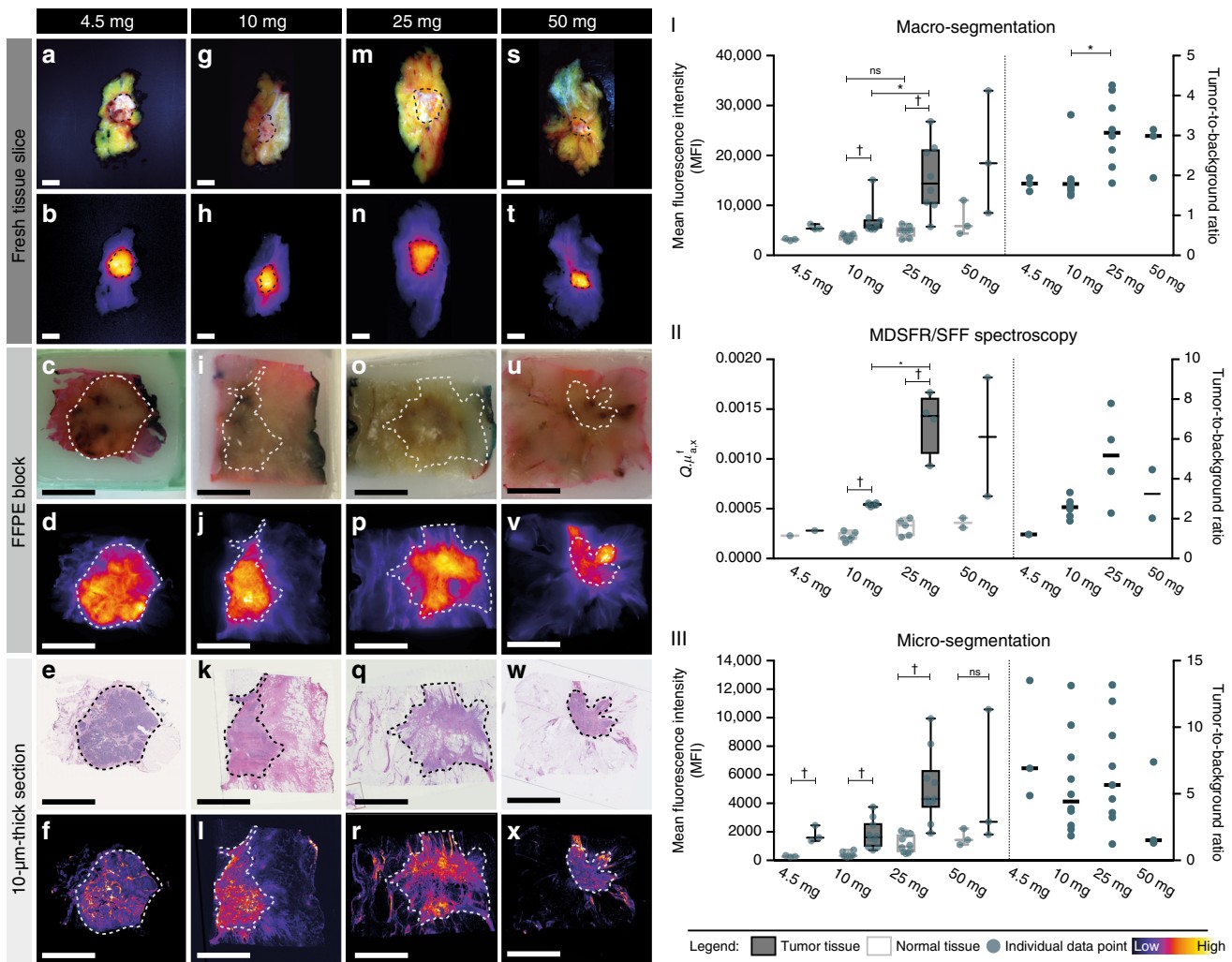

**Fig. 2** Representative images per dose group and per optical imaging method for ex vivo analyses, including MDSFR/SFF spectroscopy. Columns represent the four dose groups: 4.5 mg (**a-f**), 10 mg (**g-l**), 25 mg (**m-r**), 50 mg (**s-x**). Rows represent the imaging modality, in the upper part a white light image and in the lower part the representative fluorescence image. Tumor tissue is delineated with a dashed line. Scale bars represent 1 cm. **I** Mean fluorescence intensity (MFI) of normal tissue (gray) and tumor tissue (black) are depicted per dose group on the left *y*-axis, the right *y*-axis shows the tumor-to-background ratio per patient per dose group for macro-segmentation analyses, in **II** for MDSFR/SFF spectroscopy measurements and in **III** for micro-segmentation analyses. Fluorescence images are scaled using the most optimal minimum and maximum displayed value. Boxplot centerline is at median, the bounds of the box at 25th to 75th percentiles, the whiskers are depicting the min–max, tumor-to-background ratio data are depicted per patient; line indicates median value per dose group. Asterisk denotes significant (*P* < 0.05, Kruskal–Wallis test) values. Obelisk denotes significant (*P* < 0.05, Mann–Whitney *U*-test) values. FFPE formalin-fixed, paraffin embedded, MDSFR/SFF multi-diameter single-fiber reflectance/single-fiber fluorescence. $Q.\mu^f_{a,x}$ the product of the quantum efficiency across the emission spectrum, $Q[-]$, where $Q$ is the fluorescence quantum yield of IRDye-800CW and $\mu_{af}$ [mm$^{-1}$] is the tracer absorption coefficient at the excitation wavelength

tight macroscopic fluorescence imaging device malfunctioned; these patients were excluded. Quantitative macro-segmentation analyses confirmed significantly higher fluorescence signals in tumor tissue relative to normal background tissue in the 10 and 25 mg dose groups (Fig. 2-I). The MFI of tumor tissue increased from a median of 5368 in the 4.5 mg group to a median of 18,472 in the 50 mg group (Fig. 2-I). The 25 mg dose group showed a significantly higher MFI in tumor tissue compared to tumor tissue in the 10 mg dose group (median MFI 25 mg = 14,390, median MFI 10 mg = 6014; *P* = 0.0297, Kruskal–Wallis test). No increase in MFI of normal background tissue was observed between the 10 and 25 mg dose groups (*P* = 0.0880, Kruskal–Wallis test), resulting in a significantly higher TBR of 3.07 in 25 mg group patients versus 1.79 in 10 mg group patients (*P* = 0.0097, Kruskal–Wallis test)(Fig. 2-III).

**MDSFR/SFF spectroscopy**. MDSFR/SFF spectroscopy was performed on the fresh tissue slices in order to quantify the intrinsic fluorescence by correcting the fluorescence signal for the tissue optical properties scattering and absorption. Per patient three spots in the same fresh tissue slice were measured of both tumor tissue and normal tissue, per spot three measurements were done. In the 13 patients with available MDSFR/SFF data, intrinsic fluorescence in tumor tissue was significantly higher compared to normal tissue in the 10 mg dose group (*P* = 0.0022, Mann–Whitney *U*-test) and the 25 mg dose group (*P* = 0.0159 Mann–Whitney *U*-test) (Fig. 2-II). Furthermore, a larger variation of fluorescence intensity between patients was observed in the 25 and 50 mg groups. When comparing results of MDSFR/SFF spectroscopy with macro-segmentation of the fresh tissue slices, a similar trend of increasing fluorescence levels in tumor

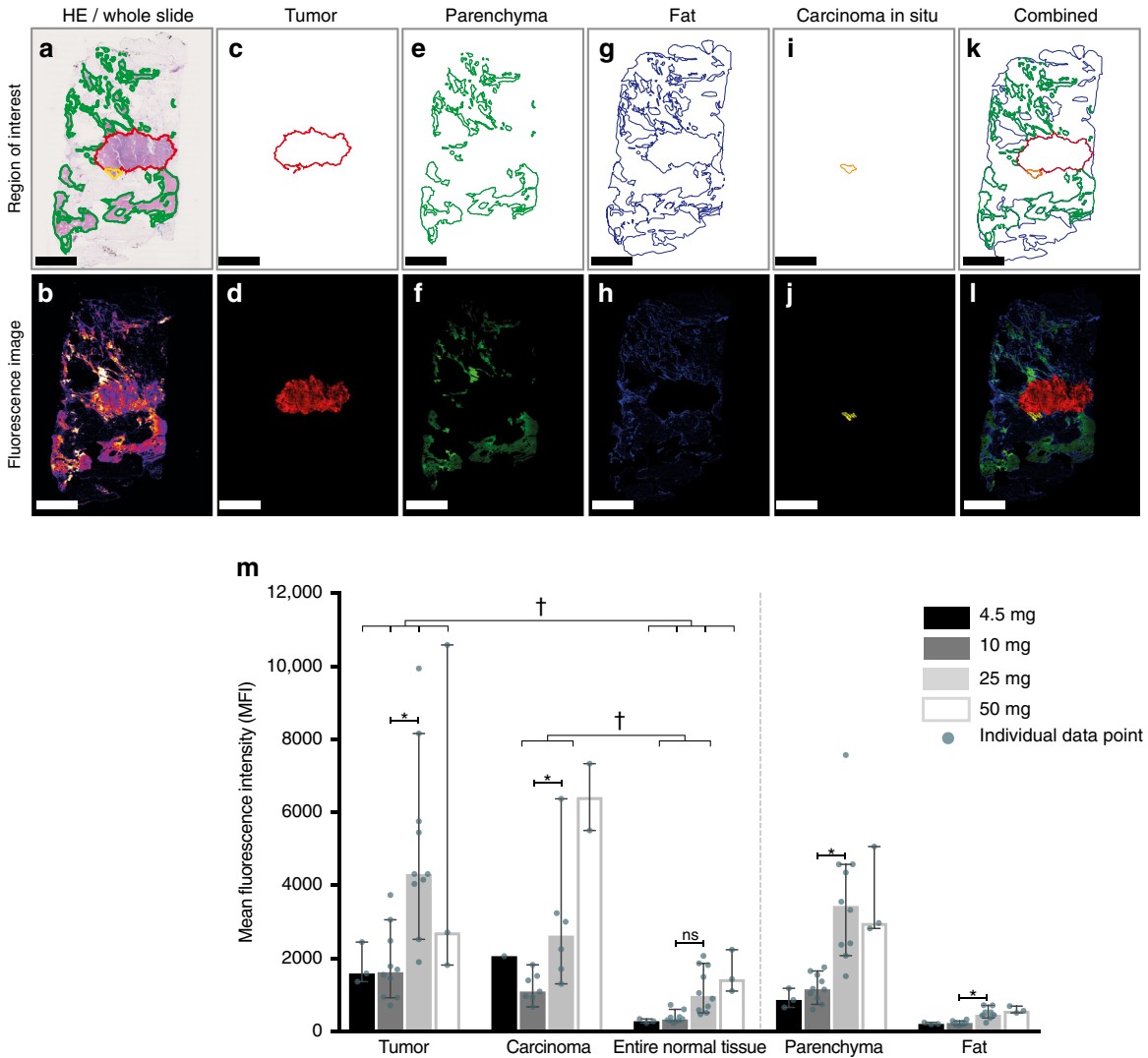

**Fig. 3** Microscopic biodistribution in breast cancer tissue of bevacizumab-800CW based on micro-segmentation analyses. The upper row shows a representative example of the region-of-interest per tissue type based on H/E staining. The lower row shows the corresponding pseudo color fluorescence intensity image of each tissue type. In **a**, **b** the whole section is depicted, and in **c**, **d** the tumor area, **e**, **f** parenchymal breast tissue including collagen, **g**, **h** fat tissue, **i**, **j** carcinoma in situ tissue, and a combination of all tissue types in **k**, **l**. Mean fluorescence intensities of all patients per dose group, per tissue type are shown in panel **m**. Asterisk denotes significant ($P < 0.05$, Kruskal–Wallis test) values. Obelisk denotes significant ($P < 0.05$, Mann–Whitney $U$-test) values. Bars are representing the median, error bars are representing 95% confidence interval. Scale bars represent 5 mm

tissue with escalating tracer doses was observed, whereas no difference in fluorescence levels was measured in background normal breast tissue between the dose groups.

**Quantitative micro-segmentation of 10-μm-thick FFPE sections**. To assess the detailed microscopic biodistribution of bevacizumab-800CW in human breast tissue, we performed micro-segmentation on a total of 200 10-μm-thick FFPE sections (Fig. 3a–l). In all 26 patients, tumor tissue showed a higher MFI compared to the entire normal breast tissue. Normal tissue was defined as fat and parenchymal breast tissue including collagen (Fig. 2-III, Fig. 3m). When analyzing the MFI per tissue type per dose group, we observed an increase in MFI for all tissue types. However, a higher tracer uptake in tumor tissue remains in escalating doses, which indicates tumor-specific targeting of the tracer irrespective of dosing (Fig. 4e–i). To visualize the differences in tumor tissue and normal parenchyma, we plotted the tumor-to-parenchyma ratio, per patient and per dose group

(median per dose group is indicated with a horizontal line). In five patients, the tumor-to-parenchyma ratio was below 1, what means that the tumor MFI was lower than the MFI of the parenchyma tissue (Fig. 4j).

**Potential clinical value of fluorescence-guided surgery**. Since macro-segmentation analyses and micro-segmentation analyses confirmed tumor-specific targeting of bevacizumab-800CW irrespective of the dosing, we evaluated the potential clinical value of molecular-fluorescence-guided surgery in breast cancer patients. We qualitatively analyzed the intraoperative fluorescence image and video data in combination with the fluorescence images of the freshly excised specimen. Intraoperative imaging took place at two time points during surgery; the tumor was imaged just before excision and the surgical cavity was imaged directly after removal of the tumor. Since this clinical trial was not designed to alter the standard of care, surgeons were not allowed to excise additional tissue based on intraoperatively detected

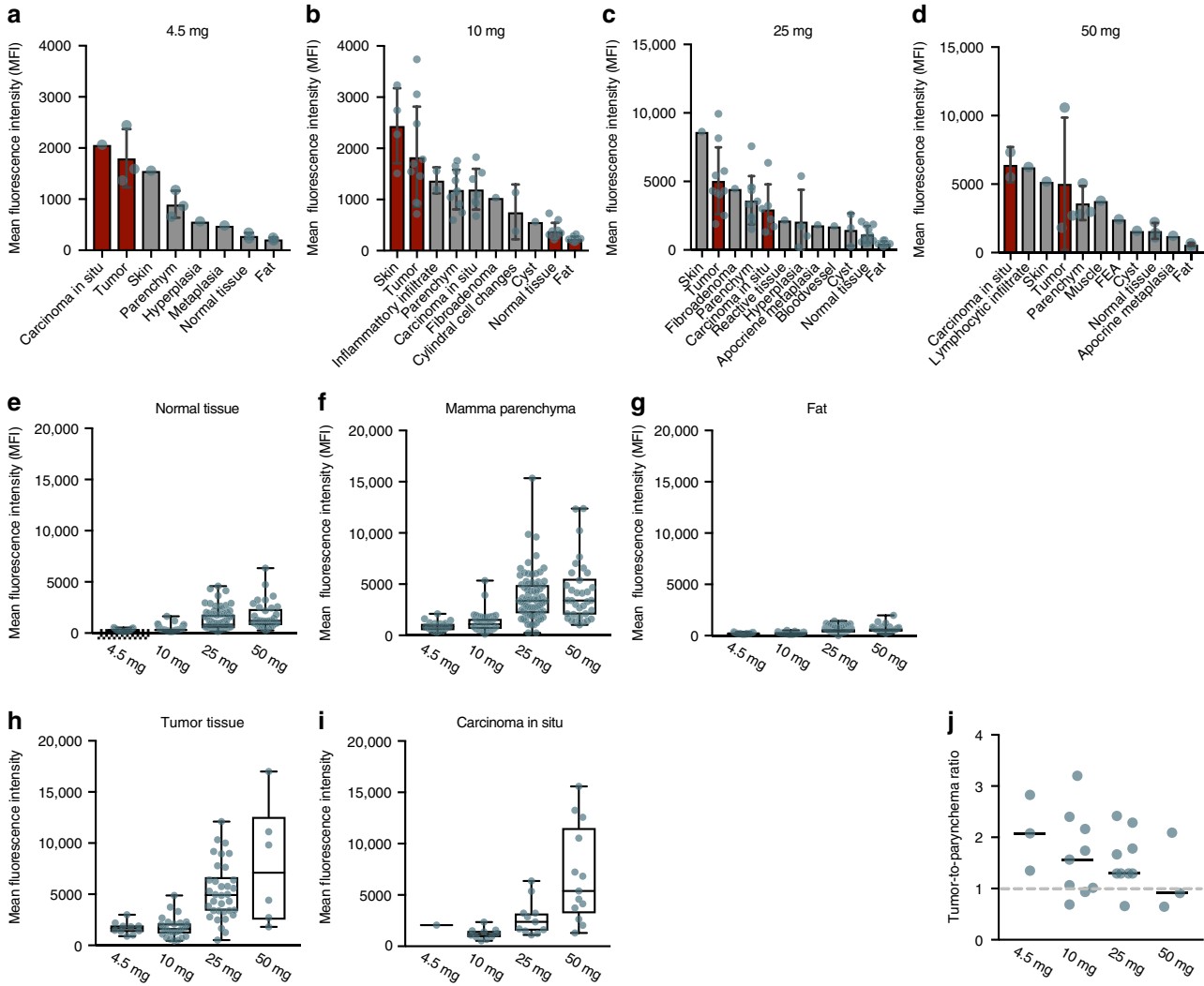

**Fig. 4** Micro-segmentation per dose group, and per tissue type. Per dose group we plotted mean fluorescence intensity per tissue type (**a**–**d**); tumor and carcinoma in situ components shown in red. The mean fluorescence intensity per tissue type was plotted in **e**-**i**. In **j** the tumor-to-parenchyma ratio per dose group is plotted. Bars represent the mean and the error bars the standard deviation. Boxplot centerline is at median, the bounds of the box at 25th to 75th percentiles, the whiskers are depicting the min–max

fluorescence signals. Therefore, intraoperative findings could only be retrospectively correlated with histopathology. Representative examples of fluorescence images from a patient with a tumor-involved surgical margin, and from a patient with a tumor-free surgical margin are presented in Fig. 5. We observed in the fluorescence scan of the 10 μm slide also non-fatty is lit up by the fluorescent tracer (Fig. 5t). We further investigated the possible cause of this high uptake by sectioning the tissue FFPE block several slides deeper, and strikingly, in these deeper sections we found tumor tissue present at the site where the high uptake is visible in the original slide (Fig. 5v). It is known that VEGF is present is in the microenvironment of the tumor[10]. Probably, the VEGF expressed in the non-fatty tissue is a field-effect from secretion from deeper seated underlying tumor cells which might explain the high bevacizumab-800CW uptake.

In eight of the 26 patients (31%) a tumor-involved surgical margin was reported after histopathological analyses; using current clinical surgical practice, none of these margins were detected intraoperatively (Table 2). When using fluorescence, in seven of these eight patients (88%) a clear fluorescence signal was detected in the surgical cavity by intraoperative fluorescence

imaging, suggesting a tumor-positive resection margin. In three of those seven patients, the primary tumor was not completely resected, whereas in four other patients the surgical margin contained additional carcinoma in situ components next to a completely resected primary tumor. In one patient, in which histopathological analysis showed ink on the invasive primary tumor, no fluorescence signal was detected in the surgical cavity. In contrast to the intraoperative imaging of this patient, ex vivo analyses of the fresh tissue slices showed clear uptake of the tracer in the tumor tissue, and also a close surgical margin was suspected on the fluorescence images of the fresh tissue slices (Supplementary Fig. 2b).

In 18/26 patients (69%) a tumor-free surgical margin was reported after histopathological analyses (Table 2). In 16 of these 18 patients (89%) no intraoperative signals were detected in the surgical cavity, whereas in the two remaining patients with a tumor-free surgical margin (2/26, 7.6%), a positive fluorescence cavity signal was detected. In these two patients, high fluorescence signals were observed in surrounding healthy tissue containing abundant collagen, normal parenchyma, accompanied by adenosis and a periductular plasma cell infiltrate as detected in

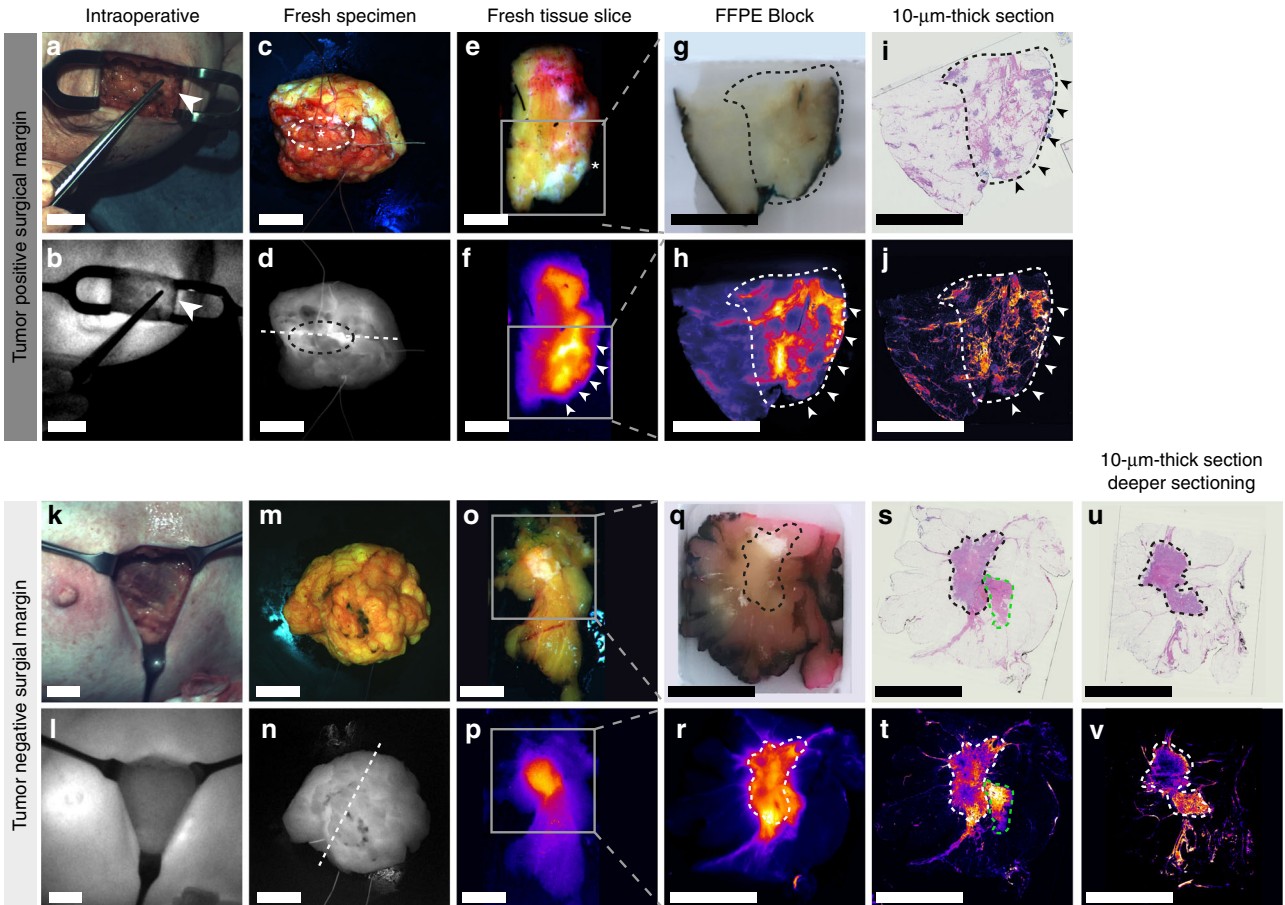

**Fig. 5** Representative examples of intraoperatively detected tumor-involved surgical margin and a tumor negative surgical margin. Columns represent from left to right intraoperative imaging, fresh specimen imaging, fresh tissue slice imaging, FFPE block imaging, and imaging of 10-μm-thick sections. The two upper rows represent a patient with a tumor-positive surgical margin, a clear fluorescence signal was detected in the surgical cavity. Subsequently, the corresponding resection plane of the excised specimen was marked with an extra suture (**a**, **b**). Fluorescence imaging of the fresh surgical specimen showed high fluorescence signals at the area of the suture mark (**c**, **d**, asterisk). Corresponding fluorescence images of fresh tissue slices, FFPE blocks and 10-μm-thick sections showed high fluorescence signals at the margin (**e**–**j**, arrows). Histopathology confirmed the presence of tumor deposits in this area (**i**). The lower rows represent a patient with a tumor-free surgical margin Fig. 5 **k**–**t**. Deeper sectioning of the FFPE block (**q**, **r**) was performed to investigate the probable cause of the high fluorescent area within the green dashed line (**t**). **u**, **v** Arrow depicts the surgical positive margin. Dashed white/black circle indicates the area with the highest fluorescence signal intensities. The asterisk represents the position of the extra suture mark. The gray box represents the origin of the FFPE block in the fresh tissue slice. The dashed white/black line delineates tumor tissue. The dashed green line delineates collagen tissue with normal parenchyma. Scale bars represent 1 cm

micro-segmentation analyses, which could explain these findings (Supplementary Fig. 2d).

## Discussion

In the emerging field of molecular fluorescence imaging a robust and broadly applicable analytical framework for clinical translation of fluorescent tracers is lacking. Based on our experience in the first clinical trials investigating fluorescence-guided surgery in human, we propose a standard evaluation methodology for clinical translation of fluorescent tracers by combining complementary qualitative and quantitative clinical optical imaging techniques[5–8].

Earlier, we demonstrated that a microdose of bevacizumab-800CW specifically targets vascular endothelial growth factor A (VEGF-A) in patients with primary breast cancer[7]. VEGF-A is present in all breast cancer types[11–14], as it is a generic target upregulated in many solid tumors and regarded one of the hallmarks of cancer[15]. Besides, we have demonstrated earlier that the antibody bevacizumab still has intact affinity for the target after conjugation with IRDye-800CW and the labeling procedure does

not influence the structural integrity and post translational modifications of bevacizumab not leading to an affected mode of action by the IRDye-800CW conjugation[16]. Data derived from preclinical studies confirm that Bevacizumab-800CW has a comparable biodistribution as [89]Zr-Bevacizumab[17].

By implementing our novel analytical framework for the first time in the current study, we confirmed the tumor-specific targeting of bevacizumab-800CW in escalating doses by tracing down bevacizumab-800CW on both a macroscopic and microscopic level within the individual components of the proposed analytical framework. Because we demonstrated the tumor-specific targeting of bevacizumab-800CW irrespective of dosing, we subsequently showed the potential clinical value of fluorescence-guided surgery in breast cancer patients, indicating the clinical feasibility and support of future studies to evaluate the definitive clinical impact of fluorescence-guided surgery in primary breast cancer patients. Using fluorescence-guided surgery in primary breast cancer patients, we showed that the intraoperative detection of tumor-involved margins is much better than standard surgical practice, this was confirmed by ex vivo analyses

| Table 2 Contingency table of molecular fluorescence-guided surgery in breast cancer patients | | | |
|---|---|---|---|
| | Surgical margin tumor positive | Surgical margin tumor negative | Total |
| Fluorescence signals in cavity positive | 7 | 2 | 9 |
| Fluorescence signals in cavity negative | 1 | 16 | 17 |
| Total | 8 | 18 | 26 |

within the analytical workflow. In seven out of eight patients, tumor-positive resection margins were detected during fluorescence-guided surgery that were missed by intraoperative assessment of surgical margins using standard visual inspection and palpation. Because the tumor-involved surgical margins could be detected intraoperatively in real time, these patients might have avoided additional surgery or therapy. This indicates the clinical value of intraoperative molecular fluorescence imaging in breast cancer patients and supports a paradigm shift in the future treatment of breast-conserving surgery, however the long-term impact of molecular fluorescence imaging on relevant clinical endpoints needs to be confirmed in next phase clinical trials, for instance the reduction in positive-margin rates (Table 2).

Besides guiding intraoperative decision making, fluorescence imaging could also have a significant impact on the workflow of pathological analysis. In current clinical practice, histological analysis of the complete surgical specimen is not possible due to practical and logistical constraints. Moreover, sampling tissue for histological analyses is based only on gross examination by visual inspection and palpation of the fresh serially sliced specimen by the attending pathologist; therefore, tumor-involved margins may not be included in total in the FFPE tissue blocks, thus causing a sampling error. Macroscopic fluorescence imaging of the fresh surgical specimen and the fresh tissue slices can provide the pathologist with a red-flag technique that precisely outlines tumor tissue (i.e., image-guided pathology). This could optimize current tissue sampling procedures and prevent sampling errors. Importantly, in our study we confirmed the cross-correlation of fluorescence-guided surgery with final histopathology, considered the gold standard. This is crucial for the further implementation of fluorescence image-guided histopathology.

Additionally, the current intraoperative clinical workflow is constrained by a considerable time lag between the clinical decision making of the surgeon (intraoperative evaluation, min-utes-hour) and the determination of presence or absence of a tumor-positive margin by the pathologist (post-operative eva-luation, days-week). We propose that image-guided pathology might bridge the gap between the surgical theatre and the pathology laboratory for reliable margin assessment, as fluores-cence images of the specimen can be provided in real-time and simultaneously to both disciplines, which will lead to a dynamic interaction between in vivo intraoperative imaging and ex vivo macroscopic imaging of the surgical specimen. This could improve surgical outcome when it counts the most—during surgery—with a direct impact on clinical decision making.

Although this study was designed as a dose escalation study, we cannot draw definitive conclusions on the optimal tracer dose for clinical decision making. This is due to the relatively small number of patients included in the lowest and highest dosing groups, leading to an unequal distribution of patients with tumor-involved surgical margins in the four dose groups. While the current study already showed the value on detecting tumor-involved surgical margins by an increased detection rate of 88%, sufficient data points are needed to determine the definitive diagnostic accuracy and to derive the optimal threshold of fluorescence intensities for intraoperative decision making.

Assuming that 15 patients with a tumor-involved surgical margin is sufficient per dose group, a total of 45–75 patients per dose group might be needed, given the 20–30% tumor-involved sur-gical margin rate in breast cancer surgery known from this study and from literature.

We observed a larger variation of fluorescence intensities in tumor tissue between patients in the 25 and 50 mg dose groups compared to 4.5 and 10 mg dose groups. Factors that might indicate protein saturation in tumors in doses from 25 mg on might be tumor size, tumor grade, and tumor type. Data derived from clinical studies evaluating cetuximab-800CW targeting Endothelial Growth Factor Receptor (EGFR) in head- and neck cancer, we learned protein saturation occurs in higher dose groups as it has shown decreasing TBRs with higher doses[18]. Based on literature data, it is known that higher VEGF mRNA expression values are associated with higher grade tumors, but also with negative ER/PR status, and positive HER2 status[19]. Most likely, the small sample size within our study limits defi-nitive conclusions about correlation of tracer uptake with clin-icopathological parameters. Therefore, the correlation of fluorescence intensity and clinicopathological parameters needs to be investigated in a next phase clinical trial.

In five patients, the tumor-to-parenchyma ratio in the micro-segmentation results was below 1, what means that the tumor MFI was lower than the MFI of the parenchyma tissue. Especially in the micro-segmentation results it becomes more apparent that the tumor-to-parenchyma ratio is lower than the TBR, compared to for example the results of the macro-segmentation analyses of the fresh tissue slices. Although parenchymal tissue including collagen showed high tracer uptake, it is to be expected that this tissue will only attribute relatively to background fluorescence intensity intraoperatively, which is supported by the fact that in only one out of these five patients this resulted in a false positive cavity signal according to the intraoperative image analyses (Supplementary Fig. 2d first row). Only large areas of par-enchymal tissue including collagen may influence the TBR in vivo, which might be challenging in patients with a tumor directly behind the nipple and in premenopausal patients with dense breasts.

We described the analytical platform which is optimized for 800 nm optical agents in particular in terms of instrumentation adapted to NIR fluorescence imaging (i.e., around the 800 nm range). Moreover, the analytical workflow is generally applicable for analyses of optical agents with other wavelengths, considering when a paired detection camera is used that is adapted to the particular wavelength of interest of the fluorophore.

In conclusion, by implementing a novel analytical workflow for molecular fluorescence imaging we have demonstrated the clin-ical feasibility of molecular fluorescence-guided surgery using the fluorescent tracer bevacizumab-800CW in breast-conserving surgery. A larger study including clinical endpoints is needed to confirm the optimal dose of bevacizumab-800CW to be used in a next phase randomized clinical trial. Furthermore, our analytical platform could be used in future clinical studies on the clinical translation and evaluation of other tumor-targeted fluorescent tracers for molecular fluorescence-guided surgery, and also in different tumor types. Therefore, this analytical platform might

serve as a standard for data collection and fluorescence image analyses in trials investigating molecular fluorescence imaging (Supplementary Fig. 5).

## Methods

**Bevacizumab-800CW synthesis.** Clinical grade bevacizumab-800CW was produced in the good manufacturing practice (GMP) facility of the UMCG by conjugating bevacizumab (Roche AG) and IRDye-800CW-NHS (LI-COR Biosciences Inc) under regulated conditions[16]. The average conjugation molecule ratio of bevacizumab (molecular weight: 149 KDa) to IRDye-800CW-NHS (molecular weight: 1.166 KDa) was 1:2, generating the conjugate bevacizumab-800CW with a total molecular weight of 151.3 KDa. Vials containing 6.0 mg bevacizumab-800CW dissolved in 0.9% sodium chloride (NaCl) solution were used to prepare the infusions in a concentration of 1 mg ml$^{-1}$. After release of the final product by the certified qualified person at the UMCG GMP facility, the tracer was intravenously administered to the subjects.

**Gel electrophoresis.** Tumor lysates of a patient from the 10 mg group, and one patient from the 25 mg group were analyzed by sodium dodecyl sulfate polyacrylamidegel electrophoresis (SDS-PAGE), to ensure the complete compound bevacizumab-IRDye800CW was present in the primary breast tumor. Additional, a lysate of normal tissue was analyzed. Results were compared with labeled and unlabeled clinically used bevacizumab. The gel was scanned with the Odyssey flatbed scanner at the 800 nm channel.

**Clinical trial design.** The dose-finding study was performed in two centers in 26 patients with proven primary breast cancer scheduled for surgery. This study was approved by the Institutional Review Board of the University Medical Center Groningen (UMCG, Groningen, the Netherlands) for conduction of the study in both the UMCG and in the Martini Hospital (MZH; Groningen, the Netherlands), a peripheral training hospital being representative for the general population of breast cancer patient operated on in The Netherlands. The study was conducted according to the principles of the Declaration of Helsinki and according to the Dutch Act on Medical Research involving Human Subjects (WMO). Patients with proven primary breast cancer scheduled for surgery were recruited during multidisciplinary breast cancer meetings in either the UMCG or Martini Hospital. Eligible patients were given orally and written information about the study and the option to participate. All human participants gave written informed consent before the start of the study procedures. An independent data safety monitoring board was appointed prior to the inclusion of the first patient to evaluate safety measures. Serious adverse events, if present, were immediately reported to the investigational review board of the UMCG, the data safety monitoring board, and the Dutch central committee on research involving human subjects (CCMO). The trial was registered at www.ClinicalTrials.gov (identifier: NCT02583568).

We designed an adapted 4 × 3 dose-finding study design, adhering to the FDA guidelines (Guidance for Industry, Developing Medical Imaging Drug and Biological Products, Part 2 Clinical Indications). This study consisted of two parts. In part I, four ascending flat doses of 4.5 mg (4.5 mL), 10 mg (10 mL), 25 mg (25 mL), and 50 mg (50 mL) bevacizumab-800CW were intravenously administered to three patients each. The dosing scheme that was used in the trial is based on the definition of microdosing. We wanted to be sure to stay more than three times below the therapeutic dose in the highest dose group. For patients who are on combination therapy with bevacizumab to treat their cancer, it is commonly accepted that the patient can safely undergo surgery 6 weeks after termination of the bevacizumab therapy: i.e., at this time the anti-angiogenetic effects have diminished sufficiently to assure there is no increased risk of bleeding or post-operative complications related to bevacizumab. The plasma levels of bevacizumab after a wash out period of 6 weeks equals the peak plasma levels after a 160 mg IV dose (as calculated by the Hospital Pharmacy and the department of Medical Oncology at the UMCG). Since the Bevacizumab-800CW will be used in surgery, the dose should stay below 160 mg total injected dose, for which the maximum flat dose of 50 mg in this clinical trial stays significantly below. We administered a flat dose per cohort, the dose was not adjusted for body weight or body surface area.

In part II, the most optimal performing dose group and one de-escalating dose were chosen on the basis of TBR to be expanded to a total of 10 subjects in each group in order to obtain a sufficient number of data points to decide on the optimal dose for a future phase III clinical study (Supplementary Fig. 1). Patients received a single dose of one of the 4 dosages bevacizumab-800CW three days prior to surgery. The lower doses of 4.5 and 10 mg were injected by slow bolus injection, and for 25 and 50 mg an infusion pump was used (infusion speed: 150 mL per hour). After injection, the infusion line was flushed with 5 mL 0.9% NaCl.

**Safety measurements.** Vital signs were measured prior to tracer injection, immediately after tracer injection and one-hour post-injection. Before tracer administration blood levels of potassium, magnesium, and calcium was measured. A pregnancy test was performed if patients were premenopausal. A standard 12-lead electrocardiogram (ECG) was made before tracer injection and one-hour post-injection. The following parameters were reported: heart rate, QT- and QTc time.

QT correction for heart rate was done using the Bazett formula. In the first 12 patients of the current study, and in 17 patients in the clinical trial NCT02113202 no QTc prolonging was observed when patients received 4,5, 10, 25, and 50 mg bevacizumab-800CW, therefore the local investigational review board and the data safety monitoring board agreed to terminate ECG measurements in Part II of this trial. Patients were asked for signs and symptoms before tracer injection, during one-hour observation period after tracer injection, and prior to surgery. After surgery, a post-surgery follow-up assessment was performed within two weeks. At this visit wound healing and adverse events were monitored.

**Standard surgical procedure.** Patients underwent either a lumpectomy ($n = 24$) or a mastectomy ($n = 2$) with or without a sentinel lymph node biopsy or axillary lymph node dissection, according to institutional standard of care procedures and guidelines. Tumor localization was done with either manual palpation, wire guidance, or using an iodine seed according to standard clinical care. Sentinel lymph node mapping was done using $^{99m}$Technetium using a gamma-probe, $^{99m}$Technetium was injected intratumorally one day before surgery conform standard clinical care.

Based on our previous experience in fluorescence imaging we adapted the standard of care minimally. We used blue non-fluorescent sterile covers in this study and avoided blue dye injection for sentinel lymph node mapping, as green color sterile covers and patent blue interfere with fluorescence signals.

**Intraoperative fluorescence imaging device.** We used a fluorescence camera system dedicated to detect IRDye-800CW-NHS (SurgVision BV 't Harde, The Netherlands). The system was configured with two LED lights for 800 nm illumination and one LED light for white light illumination. Real-time color and NIR fluorescence images and videos were acquired simultaneously with custom software at video rate. Fluorescence was detected using a highly sensitive electron-multiplying charge-coupled device (EMCCD) imaging sensor. In the color-NIR overlay images, 800 nm images were pseudo colored green. The working distance of the imaging system was 20 cm above the surgical field with a field of view of 15 cm × 15 cm, and a spatial resolution of ~2-line pairs per millimeter. For each experiment, settings were held constant on 50 ms exposure time and 300 gain; if fluorescence oversaturation occurred in higher dose groups we lowered the gain to 30 or 3 accordingly. Images and videos were recorded and stored in raw Flexible Image Transport System (FITS) format.

Before and after each surgical procedure the intraoperative camera system was calibrated using a calibration device (CalibrationDisk, SurgVision BV, The Netherlands). The device consists of a disk with round windows that can hold 8 clear polypropylene tubes of 0.65 ml (Catalog #15160, Sorenson, BioScience, Inc, Murray, U.S.A.) (Supplementary Fig. 4). The tubes were filled with 2% intralipid and two-fold increasing concentrations of bevacizumab-800CW from 1:6400 till 1:100 including one tube without tracer. The CalibrationDisk was used to test the system prior to and after surgery, whether low and high fluorescent signals could be detected from dilutional series and whether the system was functioning appropriately.

**Intraoperative imaging procedures.** This clinical trial was not designed to alter the standard of care, and surgeons were not allowed to excise additional tissue based on fluorescence signals intraoperatively detected. Therefore, intraoperative fluorescence imaging took place at two predefined time points during the surgical procedure: (1) after skin incision the tumor area was imaged just before excision of the complete surgical specimen, and (2) after removal of the specimen the surgical cavity was inspected for remaining fluorescence signals. During imaging, the surgeon was looking at a stand-alone computer monitor connected to the intraoperative imaging system. During the imaging procedures the ambient light of the surgical theater is switched off in order to prevent interaction of the ambient light with the fluorescence signals and also to have the highest sensitivity for detection of fluorescent signals during surgery, because the surgical field is also illuminated by the white light of the camera system, the surgeon can still see in real life what occurs in the surgical field. This set up did not influence the standard of care.

**Specimen handling.** After excision of the surgical specimen orientation marks were placed according to standard clinical care. A short-short suture marked the posterior side of the specimen and a long-long suture marked the nipple side of the specimen.

**Fluorescence imaging systems for ex vivo imaging.** The light-tight macroscopic fluorescence imaging device (SurgVision BV, The Netherlands) is designed for ex vivo fluorescence imaging and consists of an object table and a Complementary Metal Oxide Semiconductor (CMOS) camera which are fully shielded by a light-shielded box in order to create a dark imaging environment. The distance between the object table and the CMOS camera is fixed with a field of view of 10 cm by 10 cm. For each experiment, settings were held constant with a fluorescence exposure time of eight seconds. In two cases the light-tight macroscopic fluorescence imaging device did malfunction and the intraoperative imaging system was used for imaging of the fresh surgical specimen and fresh tissue slices in a dark

environment. Before each experiment started, the ex vivo imaging device was calibrated with the same calibration device as previously described.

The multi-diameter single-fiber reflectance/single-fiber fluorescence (MDSFR/SFF) spectroscopy system calibrates scattering signals in the reflectance spectra and provides a quantitative measurement of the NIR signal emitting from the bevacizumab-800CW tracer[20]. The MDSFR/SFF spectroscopy device was calibrated internally using a 6.6% intralipid phantom.

We used the Odyssey® CLX fluorescence flatbed scanning system (LI-COR Biosciences Inc. Lincoln, Nebraska) for detecting fluorescence in FFPE blocks and 10-μm-thick sections.

An inverted microscope (DMI6000B, Leica Biosystems GmbH, Wetzlar, Germany) was used for fluorescence microscopy with a pixel size 6.45 μm, a field of view: 120 × 120 mm. To optimize NIR visualization, the microscope was equipped with additional accessories, including a NIR LED light source ranging up to 900 nm (X-Cite 200DC, Excelitas Technologies, Waltham, MA, USA), an NIR filter set (microscope two band- pass filters 850–890 m–2p and a long-pass emission filter HQ800795LP; Chroma Technology Corp, Bellows Falls, VT, USA), a monochrome DFC365 FX fluorescence camera (1·4 M Pixel CCD, Leica Biosystems GmbH), and LAS-X software (Leica Biosystems GmbH). We used an acquisition time of 10 s for images of the 800 nm channel.

**Imaging procedures of the fresh surgical specimen**. All the procedures took place in a dark environment as much as possible, to prevent photobleaching of the tracer. The fresh surgical specimen is handled conforming current clinical practice (see also page 18). Upon arrival at the pathology department, the fresh surgical specimen was imaged in the light-tight macroscopic fluorescence imaging device on every six sides corresponding to the in vivo situation, which are anterior, posterior, medial, lateral, cranial, and caudal sides. The specimen was imaged on average of 60 min after removal of the tissue, image duration was 6 min per specimen. After freezing the whole fresh specimen in a −20 °C freezer for 15 min, the whole specimen was marked with black and blue ink, because these are non-fluorescent in the NIR range and do not interfere with the bevacizumab-800CW tracer signal. The limitation on the use of ink color did not affect the standard of care pathology practices in both institutions participating in the study. Subsequently, the fresh surgical specimen was serially sliced into 0.5-cm-thick fresh tissue slices. A photograph of all fresh tissue slices was made. Before formalin fixation, all fresh tissue slices were imaged on both sides in the light-tight macroscopic fluorescence imaging system. The fresh tissue slices were imaged on average of 180 min after removal of the tissue, image duration was 15 min for both sides of all fresh tissue slices of a patient. Furthermore, one fresh tissue slice per patient which clearly contained tumor based on gross examination, was used for MDSFR/SFF spectroscopy analysis. We placed the MDSFR/SFF spectroscopy probe on top of tumor tissue and normal tissue for quantitative measurements of NIR fluorescence. Per patient three spots were measured of both tissue types, per spot three measurements were done. Thereafter, the fresh tissue slices were fixed in formalin overnight. The next day, the pathologist macroscopically examined the specimen and selected tissue samples that were embedded in paraffin blocks and processed further for histological analyses. Tissue was embedded conforming standard clinical practice; in our institution the pathologist decides, based on visual inspection and palpation and gross examination, which tissue areas need to be embedded in FFPE blocks. This study was performed without altering the standard of care and therefore we did not influence the pathologist on selection of which tissue to be embedded in FFPE blocks. After the pathologist was finished with macroscopic selection, additional tissue samples were embedded if high fluorescence signals were detected in images of the fresh tissue slices in regions that would not have been embedded for standard clinical care. The tissue cassette numbers were marked on a printed photograph of all fresh tissue slices, to enable direct correlation between fluorescence signals in fresh tissue slice images and histology.

**Imaging procedures of formalin-fixed tissue**. All FFPE blocks of all patients were requested from the pathological department and were scanned with the Odyssey® CLX fluorescence flatbed scanning system. All FFPE blocks were scanned with the same imaging settings (wavelength: 800 nm, resolution 21 μm, quality: highest, intensity: 5).

We made 10-μm-thick tissue sections of all FFPE blocks of all patients. The 10-μm-thick sections were deparaffinized in xylene for two times five minutes each. It has been shown in an earlier clinical study executed by our group that dehydration or deparaffination in xylene steps has no effect on the presence of the compound, and no effect on the measurements of the fluorescent signals (unpublished data from clinical trial: Lamberts et al.)[7]. Thereafter, we left the slides to dry in the air in a dark environment. When dry, we imaged the slides using the Odyssey® CLX fluorescence flatbed scanning system (LI-COR Biosciences Inc.) with the same imaging settings to all slides (wavelength: 800 nm; resolution: 21 μm, quality: highest, intensity: 8). After scanning the tissue slides, we directly performed hematoxylin/eosin (H/E) staining to enable direct correlation between fluorescence signal and histology on the same slide. H/E slides were digitalized using a digital slide scanner (Hamamatsu, Japan).

**Fluorescence microscopy**. We made additional 4-μm-thick sections for microscopic assessment of the NIR signal derived from bevacizumab-800CW in order to evaluate the tracer distribution at a cellular level. The cell nuclei were counterstained with Hoechst (33258, Invitrogen, Waltham, MA, USA) The sections were mounted under a cover glass in modified Kaiser's glycerin.

**Macro-segmentation of the fresh tissue slices**. We used images of the fresh tissue slices of all 26 patients to determine the TBR per patient. TBR was defined as the MFI measured in breast cancer tissue divided by the MFI in surrounding healthy tissue at macroscopic level. We used images of the fresh tissue slices as a representative model for the in vivo situation for the macro-segmentation analyses for calculating the TBR. Fresh tissue represents the in human situation best because this tissue is not yet fixed with formalin or embedded in paraffin and the conditions of imaging are the most optimally standardized. The tumors within the slices are all on the surface without overlaying tissue, the distance from stage to camera is equal in all patients, and no ambient light is influencing the fluorescent signals. All raw (FITS-format) fluorescence images of fresh tissue slices that contained tumor tissue were imported in ImageJ (Fiji, version 1.0). ROIs were defined using the analytical workflow, because we could exactly correlate the origin of all FFPE blocks from the fresh tissue slice. As we know this origin we used the corresponding histological slice to confirm tumor areas and background areas of normal tissue in the fresh tissue slices. ROIs of the total tumor tissue area, as well as the total background tissue per fresh tissue slice are defined by MK and drawn manually. Mean fluorescence intensities (MFI, arbitrary units) of all fresh tissue slices containing tumor tissue were measured per ROI and averaged per tissue type per patient, resulting in a MFI of tumor tissue and MFI of background tissue per patient. The TBR was calculated for each patient by dividing the MFI of tumor tissue by the MFI of surrounding healthy tissue. After each dose group was finished, MFI of tumor tissue, MFI of healthy surrounding tissue and TBR for each patient were plotted in graphs (GraphPad Prism, version 7.0b). Derived from previous studies executed with 800CW labeled cetuximab[18], it was anticipated that a plateauing of TBRs level might occur with increasing doses and therefore further increasing the dose is of no further clinical need in terms of imaging TBRs. Once the TBR reached a plateau, it was considered as an indicator that the optimal dose was reached.

**MDSFR/SFF spectroscopy**. The MDSFR/SFF spectroscopy gains two reflectance spectra via two different optical fibers and one raw fluorescence spectrum. The scattering and absorption coefficients were determined from the reflectance spectra, which were used to determine the intrinsic fluorescence ($Q_f\mu^f_{a,x}$) of bevacizumab-800CW by correcting the fluorescence spectrum for the calculated tissue optical properties[20]. The intrinsic fluorescence $Q_f\mu^f_{a,x}$ is defined as the product of the quantum efficiency across the emission spectrum, $Q[-]$, where $Q$ is the fluorescence quantum yield of IRDye-800CW and $\mu_{af}$ [mm$^{-1}$] is the tracer absorption coefficient at the excitation wavelength. MDSFR/SFF spectroscopy was performed in UMCG patients only; since the system was not available in the MZH center. In one patient, the measurements failed because the device malfunctioned.

**Micro-segmentation for assessment of the biodistribution**. We performed micro-segmentation of 10-μm-thick FFPE sections to determine biodistribution of bevacizumab-800CW in human breast tissue. In summary, after fluorescence scanning and HE staining of 10-μm-thick tissue sections, an experienced breast cancer pathologist (BVDV) reviewed the histology of all slides. Different tissue components (e.g., invasive carcinoma, carcinoma in situ, benign proliferative lesions, reactive lesions, and healthy parenchymal tissue including collagen and fat) were identified and delineated manually on the digitalized HE slides. Delineated tissue components were exported as ROIs using Photoshop (Adobe Creative Cloud 2017). Fluorescence images of the 10-μm-thick sections as well as the ROIs containing different tissue components were imported in ImageJ (Fiji, version 1.0). Per ROI a fluorescence measurement was performed resulting in an MFI per tissue component per slide. Per patient a mean MFI was calculated per tissue component and plotted in a graph (Graphpad Prism, version 7.0b).

**The potential clinical value of fluorescence-guided surgery**. Clinicopathological analyses of specimens were reported conforming standard clinical care, which contained at least macroscopical description, microscopical description including tumor type, modified Bloom-Richardson grade, surgical margins, and receptor status. If present, microscopic description of carcinoma in situ was reported accordingly.

All intraoperative fluorescence images and videos of the surgical cavity of all 26 patients were reviewed by MK, a trained and experienced technical team member and blinded for histopathology. The analyses of all the images took place after the study was finished and all data were collected. Patients were divided on having presence or absence of fluorescence signals in the surgical cavity. Presence of fluorescence signals was defined as clear fluorescence signals that have higher fluorescence intensities compared to lower fluorescence intensities from background tissue, what means that high fluorescence signals could be easily delineated from lower background signals. To correlate fluorescence signals with having a tumor-involved surgical margin, a contingency table was analyzed. The surgical margin was considered to be positive if ink was present on invasive cancer

or carcinoma in situ, according to the most recent SSO-ASCO guidelines on breast cancer[9,21].

**Statistical analysis**. MFI was defined as total counts per ROI pixel area in tumor and background normal tissues. Data were tested for Gaussian distribution; none of the data was normally distributed. Fluorescence signal intensities between four different dose groups were compared using Kruskal–Wallis test with a Dunn's multiple comparison test as post hoc analyses. Fluorescence signal intensities between different tissue types within one dose group was analyzed with the Mann–Whitney $U$-test. Data are presented as boxplots with bars depicting minimum and maximum values, or median values were indicated with lines. Correlation between fluorescence intensity and clinicopathological parameters was tested with Spearman's correlation test. A two-sided $P$ value of less than 0.05 was considered significant. We used GraphPad Prism, version 7.0b Software for statistical analysis.

**Data availability**. The datasets generated during and/or analysed during the current study are available from the corresponding author on reasonable request.

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

## Acknowledgements

Wytske Boersma-van Ek for her technical assistance. Emma Smeijers, Sharon Compeer, and Oumaima Boutsa for their help in processing the FFPE tissue. Our physician assistants Clara Lemstra and Arieke Prozee for helping recruiting the patients in the UMCG and medical doctor Tessa de Vries in the Martini Hospital. Pathological assistant Lisette Jansen for implementing the workflow at the pathology department in the Martini hospital. The research leading to the results was supported by an unrestricted research grant from SurgVision BV. M.K. and G.M.V.D. reports grants from the FP-7 Framework Programme BetaCure grant no. 602812, during the study

## Author contribution

M.K. and S.-Q.Q. designed the study, performed data acquisition, analyzed, and interpreted data and drafted the manuscript. M.K. and S.-Q.Q. contributed equally to this work. M.D.L. developed the GMP production of bevacizumab-800CW and critically revised the manuscript. L.J., W.K., and J.D.V. designed the study, performed breast cancer surgeries, and supervised the study. I.K. analyzed the pathology data and critically revised the manuscript. G-J.Z. interpreted the data. D.J.R. assisted in spectroscopy data acquisition, performed the spectroscopy data analysis, and critically revised the manuscript. W.B.N. obtained funding, designed the study, and interpreted the data. A.J-S. supervised the development of GMP production of bevacizumab-800CW. B.V.D.V. was involved in histopathological analyses and micro-segmentation analyses, critically revised the manuscript. G.M.V.D. obtained funding, designed and supervised the study, interpreted data, and drafted the manuscript. All authors reviewed the final manuscript.

## Additional information

**Competing interests:** G.M.V.D. is member of the scientific advisory board of SurgVision BV. The remaining authors declare no competing interests.

