## [Peer Review File · Nature Communications]

Reviewers' comments:

Reviewer #1 (Remarks to the Author):

This manuscript aims at providing a somewhat standardized evaluation framework with which to evaluate new fluorescent tracers to accurately assess tumor margins in real time during resection surgery in breast cancer patients. Early reports in this exploding field suggest that better patient outcomes can be achieved if tumor margins can be better defined by fluorescence imaging during surgery. As many academic groups and companies are entering this field now, having a clinical trial framework that doesn't alter the current standard of care in these patients is critical.

Moreover, having a roadmap to follow for evaluating new agents will help new entities better design clinical trials that are amendable to local and federal regulatory agencies. This paper is extremely timely, well written, includes appropriate statistical analyses, is very robust and, with the supplement section, provides necessary details to aid others in adopting this methodology. Disclosures appear appropriate. There are a few technical questions and some higher level questions that should be addressed or incorporated into the discussion to improve the paper.

Minor style comments:

1. Recommend using "tissue slices" instead of "bread loaf slices" as readers will understand the slicing concept.

2. Fig 5 needs a figure key to explain the colors

Technical Questions and Concerns:

1. It is stated in the abstract that "... a novel analytical framework for the clinical translation and evaluation of tumor-targeted fluorescent tracers for molecular fluorescence imaging which can be used for a range of tumor types and with different optical tracers." As written, the methodology presented is strictly optimized for 800 nm optical agents and for using both in vivo and in vitro SurgVision cameras and bevacizumab-CW800. A statement regarding the pairing of the optical agent and detection cameras should be included in the discussion section.

2. During tissue slice preparation one wonders if the optical agent could be extracted from tissue slices during either the dehydration or deparaffination in xylene steps. If not then it should be stated as such.

3. Table 1 shows a variety of breast cancer phenotypes that were present in the trial patient cohort. Does bevacizumab target all breast cancer phenotypes? Even so, the antibody has been altered by adding CW-800 to it so a literature citation that confirms preclinical tumor targeting would be helpful.

4. Please provide clarification on the dosing scheme used in the trial. Dosing cohorts are listed as 4.5, 10, 25 and 50 mg. Is this just the stated mass injected into any sized patient or are they mg/kg bw or mg/ meter squared body surface area. It is also not possible to assess the total injection volume based on the data provided in the supplement section.

5. Quenching of the fluorescent signal within tissue is not addressed. This would likely have impact on signal intensity if tissue concentrations were high enough. If not a concern at these dose levels based on other or literature data then that should be included in the discussion as well.

6. In breast cancer patients, the cancer status of draining lymph nodes is extremely relevant to their treatment plan. Does this agent also localize in nearby metastatic lymph nodes and, if so, the potential of this agent to aid treatment planning in that scenario should also be discussed.

7. Much detail was provided in the supplement section around safety and adverse event recording but there was no mention of these findings in the manuscript. Either remove this description from the supplement or include some findings in the discussion and or conclusion section.

8. Please clarify if the surgeon is looking into a computer monitor during the procedure to see the tissue fluorescence or looking directly into the surgical field and how this might impact standard of care. Companies are working on surgical suite lighting that may make direct viewing of fluorescent tissues possible.

9. Probably beyond the scope of this paper, but it would be useful to briefly discuss the longer term impact of optically guided tumor resection on patient outcomes including recurrence, the need for repeat surgery and survival. The detection results reported in this paper are indeed impressive but the ultimate success of this approach will be based on patient outcomes. This

limitation should be included.

Reviewer #2 (Remarks to the Author):

The paper by Koller et al presents an interesting method for validating novel fluorescent tracers for in vivo fluorescence surgical guidance of tumor removal in the breast. This is an important problem, and the group is well-experienced in this area and this is an extension of their work. I very much like the authors work to follow the tracer at every step in the process from surgery to the microscope slide, and I think that describing this framework is an important and novel contribution in that regard. However, the paper is lacking in crucial technical and methodological details that make it impossible to evaluate whether the presented results support the claims that are made.

Results:

- Page 6, line 126: The statement that the presented results confirm clinical "impact" are overstated, since no interventions were performed based on the results that would impact patient care. Rather the results indicate clinical feasibility in support of future studies to evaluate impact.

- The paper is organized according to the individual assessment methods used to trace the fluorescent marker from in vivo to ex vivo. However, individual sections on i) qualitative in vivo intraoperative macroscopic imaging and ii) qualitative ex vivo imaging of the fresh whole surgical specimens is missing. The first is important, as the results of this analysis were used to estimate which patients might have avoided additional surgery had the method been used. Unfortunately, there are scant details on how exactly this was done, that are crucially needed to support the claims shown in Figure 5. Individual sections for each of these assessment methods, but more critically the first, are needed to better describe the findings and support the conclusions drawn. Specifically, critical information is missing, such as:

o Who determined that a surgical cavity image was "signal-positive" or "signal-negative" for residual tumor based on intraoperative fluorescence? Was it the surgeon, or one of the technical team? Was the person in any way trained to evaluate intraoperative fluorescence images?
o How was the determination made? Was an intensity threshold used, or was it based solely on the "feeling" of the reviewer?

o When was that determination made? During surgery, or after?

o How was the information recorded?

o Was the person who made the determination blinded to the histopathology results if the determination was made after surgery?

o In the methods, a study biopsy is mentioned that was taken of suspicious areas of the cavity that was marked with a suture by the surgeon (page 16, lines 412-414). However the results of the pathology of this study biopsy seem to not be presented. If this information exists, it could be used to confirm the in vivo imaging results and should be included. If it does not exist, please clarify the paper to clearly indicate what these study biopsies correspond to.

o Figure 4: In the slices selected (i, j, s, t) it appears that all non-fatty tissue is lit up by the marker. (s) and (t) in particular show an area of tissue surrounded completely by fat, and it is not apparent what the differences are between the circled tumor areas and adjacent non-fatty, non-malignant tissue. Also, in (e, f, o, p) it is not clear how tumor areas were identified and confirmed in the fresh bread loaf slices?

o Figure 5: I think it would be preferable to present the results of classification versus histology for in vivo imaging in the standard 2x2 table format, rather than the picture shown.

o Figure 6 seems out of place. It seems more appropriate for a review article in my opinion.

- Regarding the quantitative macro-segmentation results:

o It is not clear how the tumor and normal ROI's were determined for the fresh tissue analysis.

How did the authors accurately segment and confirm tumor and normal areas in the fresh tissue, for selection of the ROI's used in the tumor to background calculation? It is understood that they were manually segmented, but what information informed accurate manual segmentation?

- o Were the ROI's in any given fresh tissue slice defined as the "total tumor area" and "total normal area", or was some other sampling scheme used for ROI definition. If so, indicate how many ROI's were selected for each tissue slice, and how the area and location of these was determined.

- o 69 slices from 23 patients were analyzed. Were all slices containing tumor from all patients analyzed? If so, 69/23 seems low. If not, how were the slices selected per patient? For instance, were slices at the edge of the tumor which presumably contained low tumor volume also selected for analysis, or were only slices from the center of the tumor selected? If the latter is true, there could be a performance bias due to selection of slices with large tumor volume, which would not necessarily be representative of the clinical situation in which areas of involved tumor at the margin might be quite small.

- Regarding the intrinsic fluorescence quantification:

- o Please comment further on the large variation of quantitative fluorescence observed between patients in the 25 mg and 50 mg groups. It appears this could be due to low tumor specificity of the agent, which marks normal tissue with higher intensity when the agent is supplied in excess of that needed to mark tumor tissue?

- o Figure 2(II): The quantity "Quaf" is not defined, nor are its units.

- Regarding the quantitative micro-segmentation:

- o Supplementary Figure 3 should be made a primary figure.

- o Page 8, lines 183-184 states that tumor tissue displayed higher MFI compared to all normal tissue in all 26 patients. Was this true for all doses? Please clarify.

- o Page 8, lines 185-186: Carcinoma in situ did not show higher MFI than other unmentioned benign tissues such as fibroadenoma and normal parenchyma in the 25 mg group (Supplementary Fig 3c), therefore the statement does not seem to be correct, or is misleading by leaving out these tissue types from the statement.

- o Figure 2(III) and Supplementary Fig. 3 seem to indicate that tumor to normal specificity (tumor to normal background) is low or inconsistent, and seems to decrease or be more variable with increasing dose. The inability to differentiate small areas of normal parenchyma and tumor (due to low TBR when areas of interest are small) could be a significant limitation in practice and could lead to low specificity and overtreatment. The macro segmentation results are more promising, but it is not clear how much of this trend is due to the selection of tissue slices with large tumor volume with large areas of adjacent fat, thereby making the differences between tumor and normal more apparent. One wonders whether tumor could be accurately determined in a tissue slice with an equivalent area of tumor, and fibroadenoma and/or normal collagenous parenchyma.

- o Figure 3: Please indicate what the error bars signify: standard deviation or standard error?

- o Were there any significant differences between tumor and normal parenchyma? In other words, if fat tissue is removed from the "Entire normal tissue" do the significant differences remain? Clinically, it is very easy to differentiate fat from breast tissue visually.

Methods:

- Page 15, line 372: The relevance of including mastectomy patients is not clear, as typically patients with mastectomy do not suffer from positive surgical margins unless the tumor is invading the chest wall. Please comment on the rationale for inclusion of these patients in this study.

- Page 15, line 379: Patent blue dye is widely used. Please comment on whether alternate dyes for lymph node mapping can be used compatibly with your method.

- Please provide details on the resolution of the intraoperative imaging system (relevant to

sensitivity to size of residual tumor).

- Page 16, System calibration: Details on system calibration are lacking. The standard is described but not how it was used. This information is important as otherwise results of absolute mean fluorescence intensity between patients cannot be reliably interpreted.

- Page 16, lines 413-414: See previous comment regarding the existence of study biopsies collected using in vivo fluorescence guidance.

- The integration times reported for the in vivo imaging system and the ex vivo imaging system made by SurgVision are quite different (30ms versus 8 seconds). What are the differences between the devices that contribute to this?

- Page 17, lines 436-438: Please provide info on acquisition time, resolution/pixel size, field of view.

- Page 17, lines 440-447: Please provide instrument specifications rather than superlative language. Words such as "optimum NIR filter set" and "highly sensitive" camera are not informative. It is not clear what is meant by "highly sensitive LED" – line 442.

- There appears to be a typo in Page 17, line 444 when describing the filter bandpass.

- Page 18, Imaging procedures:

o Please provide an estimate of elapsed time between tissue removal and imaging (average).

o Specimens were imaged on six sides corresponding to the in vivo situation. Please indicate how orientation was maintained ex vivo and how these surfaces were registered with histology.

o Please indicate how long the process of imaging the slices took. This gets to feasibility in the intraoperative workflow.

o For MDSFR/SFF spectroscopy, were multiple measurements on the same sample taken in the same spot (technical replicates) or in different areas of the same tissue type (biological replicates)? Were fatty areas and non-fatty areas of normal tissue sampled with approximately similar frequency?

o Standard clinical practice for submission of tissue varies with institution (Page 18, line 468). Please indicate what is standard clinical practice for submission of tissue blocks at your institution explicitly. If clinical practice is not to submit every block, then please describe how submitted blocks are selected.

- Page 20, lines 502-503. "Representative of the in vivo situation" – please describe the in vivo situation that tissue slice imaging corresponds to. I can see how it would correspond to imaging of the cavity for exposed residual tumor, but it is not clear that this ex vivo model is relevant to identification of subsurface tumor in a wide local excision (i.e. guidance of resection, versus guidance of inadequacy of resection, the 2 are not the same).

- Page 20, lines 506-508. Areas of tumor and normal were outlined manually in color images of the fresh tissues. How were areas of tumor and normal confirmed at the histological level to ensure accuracy?

Reviewer #3 (Remarks to the Author):

The authors present a bevy of technical data describing the analysis of near infrared fluorescent tracers in breast tissue. The application of multiple modes of analysis to a histopathologic gold standard is to be commended, and while this small data set demonstrates the feasibility of the approach, it is unclear that it meets the level of evidence required for a proposed "paradigm shift"

in breast conservation surgery.

As a style point, the “no ink on tumor” guidelines were formulated by SSO and ASTRO, then endorsed by ASCO, but are generally referred to as the SSO-ASTRO guidelines.

Some clinical questions that should be addressed:

The use of the NIR tracer requires the limitation of certain colors in the operative field. While the color of drapes and sheets is a minor revision (though potentially costly depending on the hospital setting), the inability to use blue dye for sentinel node mapping is a departure from current standard of practice, in which dual tracer mapping is considered superior to single radionuclide mapping alone.

Additionally, do the limitations on ink color affect standard of care pathology practices for determining margin?

Many surgeons utilize separate shave margins, a practice which already reduces re-operative rates for (+) margins, and should be factored into the utility of this technology for achieving that goal.

Near-tumor breast parenchymal tissue appears to demonstrate higher fluorescent intensity, which raises concerns for additional false positives.

It is unclear who analyses the NIR imaging, whether it be the operating surgeon or an additional investigator present in theatre. Given the proposed clinical impact, there should be some discussion on operator training and interobserver variability.

While it is mentioned in methods, there is no description of any adverse peri-operative events related to the use of bevacizumab (wound healing, etc.). Also, there is no mention of the additional patient/hospital burden of a pre-operative infusion/injection three days prior to the intended operation.

Figure 5 repeats data from within the text, and Figure 6 outlines a strategy rather than original data.

Table 1 lists an abbreviation (NST) that is not used within the table. Invasive ductal adenocarcinoma is listed as such in Table 1, but its alternative “invasive carcinoma of no specific type” is used in the text (Page 5, line 108).

Table 1 also lists four patients in whom additional in situ components were identified on the margin. It is presumed that these are mutually exclusive from the four patients with a positive primary tumor margin, as there are eight total reported (+) throughout the text. It may be helpful to speak more in depth on the pathologic findings, i.e. were these contiguous to primary tumor or skip lesions.

All in all, a well done study, and worthy of further refinement and investigation, but perhaps prematurely ambitious in clinical impact.

Point-by-point response to reviewer comments on manuscript NCOMMS-18-04880-T

Reviewer #1

This manuscript aims at providing a somewhat standardized evaluation framework with which to evaluate new fluorescent tracers to accurately assess tumor margins in real time during resection surgery in breast cancer patients. Early reports in this exploding field suggest that better patient outcomes can be achieved if tumor margins can be better defined by fluorescence imaging during surgery. As many academic groups and companies are entering this field now, having a clinical trial framework that doesn't alter the current standard of care in these patients is critical. Moreover, having a roadmap to follow for evaluating new agents will help new entities better design clinical trials that are amendable to local and federal regulatory agencies. This paper is extremely timely, well written, includes appropriate statistical analyses, is very robust and, with the supplement section, provides necessary details to aid others in adopting this methodology. Disclosures appear appropriate. There are a few technical questions and some higher-level questions that should be addressed or incorporated into the discussion to improve the paper.

First of all, we would like to thank the reviewer for the constructive comment regarding the roadmap to follow for evaluating new agents and in particular the timing including the statistics, robustness and the chance of adoptability.

In the following section we have attempted to answer the technical and higher-level questions in an appropriate manner to improve the paper.

Minor style comments:

1. *Recommend using "tissue slices" instead of "bread loaf slices" as readers will understand the slicing concept.*

We have changed the wording "bread loaf slices" into the words "fresh tissue slices" throughout the whole manuscript, as recommended.

2. *Figure 5 needs a figure key to explain the colors*

As suggested by reviewer 2, we changed figure 5 into a standard 2x2 table (Table 2).

Technical Questions and Concerns:

1. *It is stated in the abstract that "... a novel analytical framework for the clinical translation and evaluation of tumor-targeted fluorescent tracers for molecular fluorescence imaging which can be used for a range of tumor types and with different optical tracers." As written, the methodology presented is strictly optimized for 800 nm optical agents and for using both in vivo and in vitro SurgVision cameras and*

bevacizumab-800CW. A statement regarding the pairing of the optical agent and detection cameras should be included in the discussion section.

We would like to thank the reviewer for this technical question. The methodology as described is indeed strictly optimized for 800nm optical agents and in particular in terms of instrumentation adapted to near-infrared fluorescence imaging (i.e. around the 800nm range). However, the analytical workflow also applies for analyses of optical agents with other wavelengths, considering when a paired detection camera is used which is adapted to the particular wavelength of interest of the fluorophore being used. The suggested statement regarding the pairing of the optical agent and detection camera is included in the discussion section (page 12 line 370-379).

- 2. During tissue slice preparation one wonders if the optical agent could be extracted from tissue slices during either the dehydration or deparaffination in xylene steps. If not then it should be stated as such.*

It has been shown in an earlier clinical study executed by our group that dehydration or deparaffination in xylene steps has no effect on the presence of the compound, and no effect on the measurements of the fluorescent signals (*unpublished data from clinical trial: Lamberts et al. Clinical Cancer Research 2016*). Moreover, the fluorescent labelled antibody remains stable and intact within the tissue (see SDS-PAGE blot, Supplemental Figure 3). Bevacizumab-800CW is bound to the soluble ligand VEGF-A and therefore extraction of the compound out of the matrix seems unlikely as such and has not been our experience in now more than 120 patients injected with the bevacizumab-800CW compound and all tissue analyses from these patients.

We have added a statement of the SDS-PAGE in the results section at page 7, line 139-147, and added Supplementary figure 3.

We have added the SDS-PAGE methods in the online methods section at page 15, line 408-414

- 3. Table 1 shows a variety of breast cancer phenotypes that were present in the trial patient cohort. Does bevacizumab target all breast cancer phenotypes? Even so, the antibody has been altered by adding CW-800 to it so a literature citation that confirms preclinical tumor targeting would be helpful.*

VEGF-A is present in all breast cancer types, as it is a generic target upregulated in many solid tumors and regarded one of the hallmarks of cancer (Hanahan, Cell 2011;144;646-74). In various types of breast cancer, it is an omnipresent target of which are:

- Adenocarcinoma of the breast (Liu, Breast Cancer Res Treat 2011)
- Lobular carcinoma of the breast (Chhieng, The Breast Journal 2003)
- Papillary breast carcinoma (Rakha, J Clin Pathol 2012)
- Mucinous (Wang, oncology letters, 2017)

We have added literature references that confirm the antibody has still intact affinity for the target after conjugation with IRDye-800CW. Also, the labeling procedure does not influence the structural integrity and post translational modifications of bevacizumab. Moreover, the mode of action is not affected by the IRDye-800CW conjugation (*Ter Weele, Eur J of Pharmaceutics and Biopharmaceutics, 2016*). Data derived from preclinical studies confirm that Bevacizumab-800CW has a comparable biodistribution as ⁸⁹Zr-Bevacizumab (*Terwisscha et al, J Nucl Med 2011*).

We have added the literature citations that were suggested by the reviewer in the discussion section at page 10-11, lines 267-284

- 4. Please provide clarification on the dosing scheme used in the trial. Dosing cohorts are listed as 4.5, 10, 25 and 50 mg. Is this just the stated mass injected into any sized patient or are they mg/kg bw or mg/ meter squared body surface area. It is also not possible to assess the total injection volume based on the data provided in the supplement section.*

We administered a flat dose per cohort, the dose was not adjusted for body weight or body surface area. Furthermore, we wanted to be sure to stay more than 3 times below the therapeutic dose in the highest dose group. For patients who are on combination therapy with bevacizumab to treat their cancer, it is commonly accepted that the patient can safely undergo surgery 6 weeks after termination of the bevacizumab therapy: i.e. at this time the anti-angiogenetic effects have diminished sufficiently to assure there is no increased risk of bleeding or post-operative complications related to bevacizumab. The plasma levels of bevacizumab after a wash out period of 6 weeks equals the peak plasma levels after a 160 mg IV dose (as calculated by the Hospital Pharmacy and the department of Medical Oncology at the UMCG). Since the Bevacizumab-800CW will be used in surgery, the dose should stay below 160 mg total injected dose, for which the maximum flat dose of 50 mg in this clinical trial stays significantly below.

The concentration of the tracer is 1 mg/mL, which means that the corresponding doses will consist of: 4.5 mL, 10 mL, 25 mL and 50 mL of the tracer.

We clarified this in the online methods section at page 16 line 441-453

- 5. Quenching of the fluorescent signal within tissue is not addressed. This would likely have impact on signal intensity if tissue concentrations were high enough. If not a concern at these dose levels based on other or literature data then that should be included in the discussion as well.*

In the dilutional series, as executed by using the CalibrationDisk, both in the lower as in the higher concentration there is no significant effect of quenching as the fluorescence intensities measured remain linear (see graph below). Therefore, the phenomenon of quenching seems very unlikely. Within tissue we also do not observe

a significant decrease in dose-escalation, nor have other groups such as the Rosenthal group in Stanford using cetuximab-IRDye800CW¹.

6. *In breast cancer patients, the cancer status of draining lymph nodes is extremely relevant to their treatment plan. Does this agent also localize in nearby metastatic lymph nodes and, if so, the potential of this agent to aid treatment planning in that scenario should also be discussed.*

We do agree with the reviewer that the lymph nodes status is very relevant in the current clinical treatment of breast cancer patients. We have investigated whether the tracer is specific for detection of metastases in lymph nodes so the tracer would serve two purposes: i) margin detection, and ii) detection of lymph node metastases.

In our cohort, 4 patients had micro-metastases in their sentinel lymph nodes, with an additional 4 other patients in which isolated tumor cells were found in their sentinel lymph node. All freshly resected lymph nodes were imaged with the back-table fluorescence imaging device. Fluorescence images were correlated with lymph node status. However, we found no correlation between fluorescence intensities and lymph node status. A quite important note is that the analysis of the lymph nodes comes with some uncertainties (see the points below), therefore we have decided not to include this in the manuscript as this might cause confusion by the readers and we decided to focus primarily on the margin detection capacity of bevacizumab-800CW and the underlying analytical platform.

Issues related to lymph node analyses

- The analysis can only be performed on fresh resected tissue (containing fatty tissue + lymph nodes), because during the pathological clinical practice this whole tissue is embedded in paraffin and in general the full thickness of the FFPE block is sectioned, which means that no tissue is left for our fluorescence analyses on a microscopic level.
- Because the analysis can only be performed on the fresh resected tissue (containing the sentinel lymph node and fatty tissue), the thickness and the size of the tissue piece does differ per node and per patient and might influence the fluorescence signal of the lymph node.
- Since we found no correlation between fluorescence intensity and lymph node status, it is not very likely that we could identify tumor positive nodes intraoperatively with this wide field technique. In the future, when for example opto-acoustic tomography can be applied to achieve higher resolution detection, it might be feasible to detect tumor positive lymph nodes as well.

7. *Much detail was provided in the supplement section around safety and adverse event recording but there was no mention of these findings in the manuscript. Either remove this description from the supplement or include some findings in the discussion and or conclusion section.*

We thank the reviewer for the comment. We added the (S)AE's recorded in the study in the patients' characteristics table. And, we added a description of the AE's reported in the results section at page 5 line 101-105.

8. *Please clarify if the surgeon is looking into a computer monitor during the procedure to see the tissue fluorescence or looking directly into the surgical field and how this might impact standard of care. Companies are working on surgical suite lighting that may make direct viewing of fluorescent tissues possible.*

Currently, the surgeon is indeed looking at a computer monitor either as a stand-alone on the imaging cart or the imaging camera is linked to monitors attached to the ceiling in the OR, which is standard in modern operating theatres as laparoscopic imaging is often standard of care in many hospitals. During the imaging procedures the ambient light of the surgical theater is switched off in order to prevent interaction of the ambient light with the fluorescence signals and also to have the highest sensitivity for detection of fluorescent signals during surgery. During the study we imaged on fixed time points to avoid too much interaction with the standard of care, however, because the surgical field is also illuminated by the white light of the camera system, the surgeon can still see in real life what occurs in the surgical field. There are already technological developments of holographic presentation of imaging data within the surgical field ², but these are very experimental which will result as of today in a high probability of non-reproducible data. Consequently, we described our analytical platform which fits seamless in the standard OR. It would be of great value to the workflow if in the future fluorescent signals could be projected into the surgical field.

We explained this in more detail on page 19, line 525-532 of the methods section.

9. *Probably beyond the scope of this paper, but it would be useful to briefly discuss the longer-term impact of optically guided tumor resection on patient outcomes including recurrence, the need for repeat surgery and survival. The detection results reported in this paper are indeed impressive but the ultimate success of this approach will be based on patient outcomes. This limitation should be included.*

We agree with the reviewer that describing the long-term impact of optically guided tumor resection on patient outcomes is beyond the scope of this paper, because this is an early phase clinical trial. However, it is relevant for future next phase clinical trials to determine which endpoints should be evaluated for image guided surgery related studies. Especially in breast cancer the survival rates and duration of patients with primary non-metastasized breast cancer are very high (currently 77% of the patients have a more than 10-year survival in The Netherlands), therefore it is not quite feasible to define disease-free-survival (DFS) or overall survival (OS) as a realistic clinical endpoint. Consequently, surrogate markers (as being used in many other disease areas like cardiovascular disease) are used which are known to be related to recurrence of disease (i.e. positive margin rate in breast cancer). A future clinical trial should therefore include the number of positive margins with and without fluorescence guided surgery (FGS). It is questionable whether a clinical trial can ever be blinded due to the nature of the procedure and presentation of imaging data, but at least standard-of-care alone should be compared by standard-of-care combined with molecular fluorescence guided surgery.

A statement is included in the discussion section on page 11 line 301-303.

Reviewer #2 (Remarks to the Author):

The paper by Koller et al presents an interesting method for validating novel fluorescent tracers for in vivo fluorescence surgical guidance of tumor removal in the breast. This is an important problem, and the group is well-experienced in this area and this is an extension of their work. I very much like the authors work to follow the tracer at every step in the process from surgery to the microscope slide, and I think that describing this framework is an important and novel contribution in that regard. However, the paper is lacking in crucial technical and methodological details that make it impossible to evaluate whether the presented results support the claims that are made.

First of all, we would like to thank the reviewer for the positive remarks on the described framework and indicating the importance and novelty of this framework. We also would like to thank the reviewer for the technical and methodological questions.

In the following section we have attempted to answer the technical and methodological questions in an appropriate manner to improve the paper.

Results:

- *Page 6, line 126: The statement that the presented results confirm clinical “impact” are overstated, since no interventions were performed based on the results that would impact patient care. Rather the results indicate clinical feasibility in support of future studies to evaluate impact.*

We agree with the reviewer that the sentence: “the presented results confirm clinical impact” might be too much of an overstatement. We rephrased this sentence into the reviewer’s suggestion: “the results indicate clinical feasibility in support of future studies to evaluate impact”

- *The paper is organized according to the individual assessment methods used to trace the fluorescent marker from in vivo to ex vivo. However, individual sections on i) qualitative in vivo intraoperative macroscopic imaging and ii) qualitative ex vivo imaging of the fresh whole surgical specimens is missing idem. The first is important, as the results of this analysis were used to estimate which patients might have avoided additional surgery had the method been used. Unfortunately, there are scant details on how exactly this was done, that are crucially needed to support the claims shown in Figure 5. Individual sections for each of these assessment methods, but more critically the first, are needed to better describe the findings and support the conclusions drawn.*

We thank the reviewer for the comments regarding the description of the methods. We therefore improved several parts of the methods section to be clearer on items described above and below.

Specifically, critical information is missing, such as:

- *Who determined that a surgical cavity image was “signal-positive” or “signal-negative” for residual tumor based on intraoperative fluorescence? Was it the surgeon, or one of the technical team? Was the person in any way trained to evaluate intraoperative fluorescence images?*

MK, medical doctor and a technical team member, determined the negativity or positivity of a fluorescent signal. She is fully trained and experienced to evaluate fluorescence images. MK was blinded for pathology results. This is clarified in the online methods section on page 25 line 740-742.

- *How was the determination made? Was an intensity threshold used, or was it based solely on the “feeling” of the reviewer?*

We used intraoperatively the same camera settings in all patients (exposure time, gain, distance to the surgical field), and with the current camera systems it is possible to quantify fluorescent signals during surgery or intraoperatively by arbitrary units. Therefore, in our experience, trained team members can evaluate fluorescent signals as positive or negative. The

evaluation is based on both the signal intensities and the fluorescence distribution in the images.

This is clarified in the online methods section on page 25, line 744-745

- *When was that determination made? During surgery, or after?*

The images were analyzed after they were obtained from all 26 patients. This is clarified on page 25 line 741-742.

- *How was the information recorded?*

Raw images and videos (FITS format) were collected from each patient on two different time points; after incision and approaching the tumor, and after resection. As described on page 18 line 505-506.

- *Was the person who made the determination blinded to the histopathology results if the determination was made after surgery?*

The person (MK) that determined the fluorescence were blinded for the histopathological results, as clarified in an earlier response. This was clarified in the manuscript on page 25 line 740

- *In the methods, a study biopsy is mentioned that was taken of suspicious areas of the cavity that was marked with a suture by the surgeon (page 16, lines 412-414). However, the results of the pathology of this study biopsy seem to not be presented. If this information exists, it could be used to confirm the in vivo imaging results and should be included. If it does not exist, please clarify the paper to clearly indicate what these study biopsies correspond to.*

We do agree with the reviewer that the results of the biopsies taken of suspicious areas were not reported. One should notice that biopsies taken from suspicious areas were not taken based on a particular threshold or cutoff of fluorescence signal intensities but merely on a subjective judgement by the research team. Also, the distribution of the biopsies taken per dosing group was highly variable with different numbers per dosing group. As such, the interpretation of the final histopathology results of the biopsies taken was impossible to correlate to a certain fluorescence intensity with either a tumor-positive or tumor-negative biopsy. In our opinion, this part within the methods section should therefore be omitted and left out of the reporting of the data due to a limited number of data points.

- *Figure 4: In the slices selected (i, j, s, t) it appears that all non-fatty tissue is lit up by the marker. (s) and (t) in particular show an area of tissue surrounded completely by fat, and it is not apparent what the differences are between the circled tumor areas and adjacent non-fatty, non-malignant tissue. Also, in (e,*

f, o, p) it is not clear how tumor areas were identified and confirmed in the fresh bread loaf slices?

We do partly agree with the reviewer about the statement that all non-fatty tissue is lit up by the marker. The image in Panel J shows indeed uptake of the tracer in non-fatty tissue, however, the tumor tissue within the marked area showed much higher fluorescence signals (yellow – white), compared to the fluorescence intensities of the tissue in the non-marked area (red).

We do agree with the reviewer that in the 10 μm slide in panel t of figure 4, also non-fatty is lit up by the fluorescent tracer as detected by the flatbed-scanner. We further investigated the possible cause of this high uptake by deeper sectioning the tissue FFPE block several slides deeper, and strikingly, in these deeper sections we found tumor tissue present at the site where the high uptake is visible in the original slide. It is known that VEGF is present in the microenvironment of the tumor³. Probably, the VEGF expressed in the non-fatty tissue is a field-effect from secretion from deeper seated underlying tumor cells which explains the high bevacizumab-800CW uptake. Interestingly, when comparing fluorescence intensities of normal tissue adjacent to tumor tissue (i.e. normal breast tissue in a slide containing tumor) to normal tissue in non-tumor slides, a higher MFI was observed in adjacent normal tissue as compared to normal tissue in non-tumor slides in all dose groups (see graph below). Although, the sample size was too small to find any statistical differences.

We have added the additional slides of the deeper cutting to the figure (panel u and v) and explained the high uptake as noticed by the reviewer in the text (page 9, line 229-236).

- *Figure 5: I think it would be preferable to present the results of classification versus histology for in vivo imaging in the standard 2x2 table format, rather than the picture shown.*

We have replaced the original Figure 5 by a standard 2x2 table format (**Table 2**).

- *Figure 6 seems out of place. It seems more appropriate for a review article in my opinion.*

Although the Figure 6 might be more appropriate for a review article to the reviewers' opinion, we do believe that this figure explains the position of the paper in future evaluations of fluorescent tracers in different phases of clinical trials. We positioned this figure as supplemental.

Regarding the quantitative macro-segmentation results:

- *It is not clear how the tumor and normal ROI's were determined for the fresh tissue analysis. How did the authors accurately segment and confirm tumor and normal areas in the fresh tissue, for selection of the ROI's used in the tumor to background calculation? It is understood that they were manually segmented, but what information informed accurate manual segmentation?*

The selection of ROIs for tumor and background were determined by two steps:

1. The white-light images of the fresh tissue slices give an indication of the location of the tumor and background tissue (as is currently also the standard-of-care executed by pathologists in routine examination of the specimen)
2. Since we know the original location of the tissue in all FFPE blocks, the corresponding histological slice was used to confirm whether this area was histopathologically proven tumor tissue.
3. The ROI was manually drawn at the corresponding fluorescence image of the fresh tissue slice.

To clarify: fresh tissue slices are about 5mm thick, whereas a histological slice is 4 μm thick (factor 100 difference). This implies by definition that 1 histology slide does not reflect the histology for the total thickness of the fresh tissue slice. Consequently, it is not accurate to directly overlay histology slides with fresh tissue slices to determine the ROI for tumor and background and for that reason we developed a standardized process.

These steps described above are clarified in the manuscript on page 24 line 676-688.

- *Were the ROI's in any given fresh tissue slice defined as the "total tumor area" and "total normal area" or was some other sampling scheme used for ROI definition. If so, indicate how many ROI's were selected for each tissue slice, and how the area and location of these was determined.*

The ROIs were defined as total tumor tissue area and total normal tissue area. We did not use more than one ROI per fresh tissue slice per tissue type. This is clarified in the manuscript on page 24 line 686.

- *69 slices from 23 patients were analyzed. Were all slices containing tumor from all patients analyzed? If so, 69/23 seems low. If not, how were the slices selected per patient? For instance, were slices at the edge of the tumor which presumably contained low tumor volume also selected for analysis, or were only slices from the center of the tumor selected? If the latter is true, there could be a performance bias due to selection of slices with large tumor volume, which would not necessarily be*

representative of the clinical situation in which areas of involved tumor at the margin might be quite small.

We confirm that we analyzed all fresh tissue slices containing tumor from all patients, and we did not select slices per patient. Sixty-nine fresh tissue slices derived from 23 patients means per patient a mean of three fresh tissue slices contains tumor. As one slice is about 5mm thick, so a total volume of fresh tissue slices per patient containing tumor is approximately 15mm. The mean tumor size of our patient cohort is: 14.4mm, so actually the number of the fresh tissue slices matches perfectly with the sizes of the tumor. The reviewer might have interpreted the type of slices different compared to histology slides.

Because we analyzed all slices containing tumor from all patients, we also included the edges from the tumor with less volume into the analyses. Consequently, we have included all slices in the analyses in order to prevent potential performance bias the reviewer is also mentioning.

Regarding the intrinsic fluorescence quantification:

- *Please comment further on the large variation of quantitative fluorescence observed between patients in the 25 mg and 50 mg groups. It appears this could be due to low tumor specificity of the agent, which marks normal tissue with higher intensity when the agent is supplied in excess of that needed to mark tumor tissue?*

We do agree with the reviewer that the phenomenon of a larger variation in 25mg and 50mg is very interesting. However, as the Tumor-to-Background ratios also raises with higher dose levels, it is not very likely that normal background tissue is saturated more with higher dose levels as the reviewer suggests. If that would be the case, an equal or lower Tumor-to-background ratio is to be expected, which was not the case.

We do have two hypotheses why in the 25mg and 50mg group the variation in fluorescence uptake is higher, especially the fluorescence intensity in the tumor tissue.

1. Fluorescence intensities might be more dependent on tumor size in higher dose groups, which might indicate that smaller tumors are fully saturated with the tracer whereas larger tumors still can take up tracer. Whereas in lower doses (4.5mg and 10mg) most likely none of the tumors can get saturated with the tracer, resulting in the same concentration of the tracer throughout the tumor.
2. Fluorescence intensities might be dependent on pathological characteristics, as grade, estrogen receptor positivity, progesterone receptor positivity or HER2 receptor positivity, which are also related to VEGF expression.

Factors that might indicate protein saturation in tumors in doses from 25mg on might be tumor size, tumor grade and tumor type. Data derived from clinical studies evaluating cetuximab-800CW targeting Endothelial Growth Factor Receptor (EGFR) in head- and neck cancer, we learned protein saturation occurs in higher dose groups as it has shown decreasing tumor-to-background ratios with higher doses ¹. Based on literature data, it is known that higher VEGF mRNA expression values are associated with higher grade tumors, but also with negative ER/PR status, and positive HER2

status⁴. Most likely, the small sample size within our study limits definitive conclusions about correlation of tracer uptake with clinicopathological parameters. Therefore, the correlation of fluorescence intensity and clinicopathological parameters needs to be investigated in a next phase clinical trial.

We added this line of reasoning in the discussion section on page 13, line 348-357

- *Figure 2(II): The quantity “ $Q\mu_{af}$ ” is not defined, nor are its units.*

The intrinsic fluorescence $Q\cdot\mu_{a,x}^f$ is defined as the product of the quantum efficiency across the emission spectrum, $Q[-]$, where Q is the fluorescence quantum yield of IRDye-800CW and μ_{af} [mm^{-1}] is the tracer absorption coefficient at the excitation wavelength.

We have added the described section to the methods of the MDFSR/SFF spectroscopy on page 24 line 703-707.

Regarding the quantitative micro-segmentation:

- *Supplementary Figure 3 should be made a primary figure.*

We are happy to include the supplementary figure 3 as a primary figure in the manuscript.

- *Page 8, lines 183-184 states that tumor tissue displayed higher MFI compared to all normal tissue in all 26 patients. Was this true for all doses? Please clarify.*

We confirm that in all patients, among all doses, the tumor showed higher MFI in the tumor compared to normal tissue. This is reflected by the Tumor-to-Background ratios per patient depicted in figure 2. All patients had a TBR >1 in all analyses methods.

- *Page 8, lines 185-186: Carcinoma in situ did not show higher MFI than other unmentioned benign tissues such as fibroadenoma and normal parenchyma in the 25 mg group (Supplementary Fig 3c), therefore the statement does not seem to be correct, or is misleading by leaving out these tissue types from the statement.*

The reviewer is right, this statement is misleading in this form. Therefore, we have deleted this sentence from the manuscript.

- *Figure 2(III) and Supplementary Fig. 3 seem to indicate that tumor to normal specificity (tumor to normal background) is low or inconsistent and seems to decrease or be more variable with increasing dose. The inability to differentiate small areas of normal parenchyma and tumor (due to low TBR when areas of interest are small) could be a significant limitation in practice and could lead to low specificity and overtreatment. The macro segmentation results are more promising, but it is not clear how much of this trend is due to the selection of tissue slices with large tumor*

volume with large areas of adjacent fat, thereby making the differences between tumor and normal more apparent. One wonders whether tumor could be accurately determined in a tissue slice with an equivalent area of tumor, and fibroadenoma and/or normal collagenous parenchyma.

We agree with the reviewer. Especially in the micro-segmentation results it becomes more apparent that the difference in fluorescence intensity per tissue type becomes less clear, compared to for example the results of the fresh tissue slices. However, we do not completely agree with the reviewer that this low specificity will lead directly to overtreatment. Although parenchymal tissue including collagen showed high tracer uptake, it is to be expected that parenchymal tissue including collagen will only attribute relatively to background fluorescence intensity *intraoperatively* and will not complicate *in vivo* surgical decision making, because this tissue type is mainly located in small streaks within large amounts of fat. Only large areas of parenchymal tissue including collagen may influence the tumor-to-background ratio *in vivo*, which might be challenging in patients with a tumor directly behind the nipple and in premenopausal patients with dense breasts. An example in which it is difficult to discriminate tumor tissue from nipple tissue is visualized in the bottom row of Supplementary figure 2, in which results of a patient with a tumor directly behind the nipple are depicted. A previous PET imaging study in breast cancer patients with 5mg ⁸⁹Zr-Bevacizumab also showed high tracer uptake in nipple tissue, which is likely to be due to high vascularization of the nipple, compared with normal breast tissue ⁵.

We have included this line of reasoning in the discussion section on page 13, line 360-369.

- *Figure 3: Please indicate what the error bars signify: standard deviation or standard error?*

The bar graph in Figure 3 depicts the median value and the error bars signify the 95% confidence interval as stated in the figure legend.

- *Were there any significant differences between tumor and normal parenchyma? In other words, if fat tissue is removed from the “Entire normal tissue” do the significant differences remain? Clinically, it is very easy to differentiate fat from breast tissue visually.*

To visualize the differences in tumor tissue and normal parenchyma, we plotted the tumor-to-parenchyma ratio, per patient and per dose group (median per dose group is indicated with a horizontal line). In 5 patients the tumor-to-parenchyma ratio was below 1, what means that the tumor MFI was lower than the MFI of the parenchyma tissue. However, in only one out of these five patients this resulted in a false positive cavity signal according to the intraoperative image analyses (see supplementary figure 2, D first row). This can be explained that parenchymal tissue most likely will only attribute relatively to the background signal, as described in the answer on the question at the top of this page.

To clarify: the tracer bevacizumab-800CW will not be used for intraoperative discrimination of fat from breast tissue, as it will not lead to clinical impact in which we support the opinion of the reviewer.

We have clarified this in the discussion section of the manuscript, on page 13, line 358-369.

Methods:

- *Page 15, line 372: The relevance of including mastectomy patients is not clear, as typically patients with mastectomy do not suffer from positive surgical margins unless the tumor is invading the chest wall. Please comment on the rationale for inclusion of these patients in this study.*

We agree with the reviewer that in routine clinical care there is no relevance for adding this technique during mastectomy procedures. The primary endpoint of the study was: 1) does the tracer accumulates in tumor tissue, and 2) dose-finding as based on the most optimal tumor-to-background ratio. Therefore, it was decided not only to include patients eligible for breast-conserving surgery but also patients undergoing a mastectomy.

- *Page 15, line 379: Patent blue dye is widely used. Please comment on whether alternate dyes for lymph node mapping can be used compatibly with your method.*

In our study, which is also more common practice in the Netherlands, the sentinel lymph node is intraoperatively detected solely using ^{99m}Techetium using a gamma-counter, which is also standard clinical care in The Netherlands. ^{99m}Techetium does not interfere with fluorescence imaging. We clarified this in the method section on page 17 of the manuscript.

- *Please provide details on the resolution of the intraoperative imaging system (relevant to sensitivity to size of residual tumor).*

The spatial resolution of the intraoperative camera system is approximately 2-line pairs/ millimeter at a typical working distance of 24cm. While this theoretically should allow the detection of submillimeter fluorescent targets, the ability to detect small residual tumor parts will be dependent on the amount of fluorescent agent uptake in that tissue compared to the adjacent background tissue.

We have added these technical details in the methods section on page 18, line 502-503

- *Page 16, System calibration: Details on system calibration are lacking. The standard is described but not how it was used. This information is important as otherwise results of absolute mean fluorescence intensity between patients cannot be reliably interpreted.*

We agree with the reviewer that the information on how the calibration device was used is not totally clear in the manuscript. We used the disk to test the system prior to and after surgery, whether it was able to detect the dilutional series varying from low and high fluorescent signals and whether the system was functioning appropriately. The same accounts for the specimen imaging device, which was also technically tested by using the disk prior to use. We described the use of the disk in the methods section of the manuscript (page 18, line 516-518).

- *Page 16, lines 413-414: See previous comment regarding the existence of study biopsies collected using in vivo fluorescence guidance.*

We would like to refer to our answer on the previous comment on page 8 of this document.

- *The integration times reported for the in vivo imaging system and the ex vivo imaging system made by SurgVision are quite different (30ms versus 8 seconds). What are the differences between the devices that contribute to this?*

The primary reason for the difference in integration times between the in-vivo and ex-vivo imaging is that the ex-vivo specimens are not moving and can therefore be imaged with longer exposure times, whereas the in-vivo tissue during surgery moves because of respiration, heart beat and manipulation of the surgical field by the surgeon, making short integration times necessary to avoid motion blur. As such, the intraoperative imaging system utilizes a highly sensitive EMCCD imaging sensor to allow for short integration times, whereas the ex-vivo imaging was performed using a CMOS sensor.

These technical details were added in the manuscript on page 18 lines 499-500 and on page 20, lines 554-555.

- *Page 17, lines 436-438: Please provide info on acquisition time, resolution/pixel size, field of view.*

Pixel size = 6.45 μm

acquisition time = 10 seconds for imaging in the 800nm channel.

Field of view = 120 μm x 120 μm

We added these numbers accordingly in the method section of the manuscript on page 20-21.

- *Page 17, lines 440-447: Please provide instrument specifications rather than superlative language. Words such as “optimum NIR filter set” and “highly sensitive” camera are not informative. It is not clear what is meant by “highly sensitive LED” – line 442.*

We deleted the words “optimum NIR filter set” and “highly sensitive camera”.

- *There appears to be a typo in Page 17, line 444 when describing the filter bandpass.*

The reviewer is right, ‘90’ must be ‘890’. We have changed this accordingly.

Page 18, Imaging procedures:

- *Please provide an estimate of elapsed time between tissue removal and imaging (average).*
 - The surgical cavity was imaged directly after removal of the tissue and took on average 2 minutes per patient.
 - The specimen was imaged on average 60 minutes after removal of the tissue in 6 minutes to image per specimen.
 - The fresh tissue slices were on average imaged 180 minutes after removal of the tissue in 15 minutes per patient.

We have addressed the elapsed times in the methods section on page 21.

- *Specimens were imaged on six sides corresponding to the in vivo situation. Please indicate how orientation was maintained ex vivo and how these surfaces were registered with histology.*

The specimen was processed according to standard clinical care. In the two participating hospitals in our study, a short-short suture mark was placed on the posterior side, and a long-long suture mark on the nipple-side of the specimen. All specimen were transferred to the pathology department accompanying with a digital letter in which the surgeon indicates the exact location of the tumor. The pathologist

reconstructs the orientation of the specimen in the breast. All study personnel were up to date about the exact location of the tumor in the breast and informed about the standard clinical procedures for orientation marks. This enabled reconstruction of the orientation ex-vivo to the in-vivo one.

This is written on page 19 under the heading 'specimen handling'.

After the specimen was imaged, the specimen was inked before fresh tissue slices were cut, to enable correlation of ink color with the histological slides. (also, standard clinical procedure).

This is written on page 21 of the manuscript, lines 598-602.

- *Please indicate how long the process of imaging the slices took. This gets to feasibility in the intraoperative workflow.*

We do agree with the reviewer that it is very interesting to add the imaging of the fresh tissue slices into the intraoperative workflow, and it would be very helpful to investigate the feasibility in a next phase clinical trial. In the current clinical trial, it is not used for intraoperative clinical decision making, but within the analytical platform the images of the fresh tissue slices were used for quantification and determination of the tumor specific targeting of the tracer. Because it was an early phase trial, we deliberately worked-out an imaging procedure that would fit seamless in the current standard clinical procedure. In the future one could imagine that the specimen will be sliced in the OR and fluorescently imaged for supporting 'real-time' intraoperative decision making, taking into account an efficient and smooth workflow of imaging the specimen on the back-table.

The process of imaging all the fresh slices took an average of 15 minutes per patient. However, we must admit that the current camera systems already made quite some progress in imaging speed. Currently, we use the Explorer Vault imager of SurgVision, which enables imaging of 4 fresh tissue slices per image in 50 milliseconds, which makes back-table imaging in the OR a realistic option to carry out in future studies.

- *For MDFSFR/SFF spectroscopy, were multiple measurements on the same sample taken in the same spot (technical replicates) or in different areas of the same tissue type (biological replicates)? Were fatty areas and non-fatty areas of normal tissue sampled with approximately similar frequency?*

We performed technical replicates and also biological replicates. Three measurements of the same spot were imaged, and per tissue types 2-3 different areas were measured.

Non-fatty areas of normal tissue were not feasible to measure because these tissues were too small for putting the fiber on.

We adjusted the methods for the acquisition of the MDFSFR/SFF spectroscopy measurements to clarify this. (page 21, line 610-611)

- *Standard clinical practice for submission of tissue varies with institution (Page 18, line 468). Please indicate what is standard clinical practice for submission of tissue blocks at your institution explicitly. If clinical practice is not to submit every block, then please describe how submitted blocks are selected.*

In our institutions it is not common practice to embed the whole surgical specimen in FFPE blocks. The pathologists decide, based on visual inspection and palpation, which tissue areas need to be embedded in FFPE blocks. This study was performed without altering the standard of care and therefore we did not influence the pathologist on selection of which tissue to be embedded in FFPE blocks. After the pathologist was finished with macroscopic selection, additional tissue samples were embedded if high fluorescence signals were detected in images of the fresh tissue slices in regions that would not have been embedded for standard clinical care. Because this standard clinical practice in pathology may vary considerably between institutions, it is very important to have standardization and uniform handling of the specimen. Therefore, appropriate and sufficient training and the use of fluorescent image-guided pathology may assist in this in the near future.

This is clarified in the manuscript on page 22, lines 624-629.

- *Page 20, lines 502-503. "Representative of the in vivo situation" – please describe the in vivo situation that tissue slice imaging corresponds to. I can see how it would correspond to imaging of the cavity for exposed residual tumor, but it is not clear that this ex vivo model is relevant to identification of subsurface tumor in a wide local excision (i.e. guidance of resection, versus guidance of inadequacy of resection, the 2 are not the same).*

We do agree with the reviewer that this sentence is somewhat confusing. What we meant is the following: we analyzed the TBR in the fresh tissue slices because this is the most representative model as the tissue is fresh and not fixed with formalin or embedded in paraffin, and the conditions of imaging are the most optimally standardized. The tumors within the slices are all on the surface without overlaying tissue, the distance from stage to camera is equal in all patients, and no ambient light is influencing the fluorescent signals.

We have clarified this in the manuscript on page 23, lines 669-675

- *Page 20, lines 506-508. Areas of tumor and normal were outlined manually in color images of the fresh tissues. How were areas of tumor and normal confirmed at the histological level to ensure accuracy?*

We would like to refer to the first question of this reviewer on the quantitative macro-segmentation results on page 10 of this document.

Moreover, by performing the whole analytical workflow we know from which location in the fresh tissue slice the FFPE blocks originate, which is exactly the benefit of executing the analytical workflow.

Reviewer #3 (Remarks to the Author):

The authors present a bevy of technical data describing the analysis of near infrared fluorescent tracers in breast tissue. The application of multiple modes of analysis to a histopathologic gold standard is to be commended, and while this small data set demonstrates the feasibility of the approach, it is unclear that it meets the level of evidence required for a proposed “paradigm shift” in breast conservation surgery.

We would like to thank the reviewer for the constructive comments that definitely will improve the paper.

As a style point, the “no ink on tumor” guidelines were formulated by SSO and ASTRO, then endorsed by ASCO, but are generally referred to as the SSO-ASTRO guidelines.

Thank you for this comment, we changed the wording accordingly.

Some clinical questions that should be addressed:

- *The use of the NIR tracer requires the limitation of certain colors in the operative field. While the color of drapes and sheets is a minor revision (though potentially costly depending on the hospital setting), the inability to use blue dye for sentinel node mapping is a departure from current standard of practice, in which dual tracer mapping is considered superior to single radionuclide mapping alone.*

We do understand the complexity of NIR imaging in relationship to imaging of visible light and invisible near-infrared light. Therefore, we would like to explain the concept in relationship to colors and the concept of autofluorescence. As the camera system consists of two separate cameras, one for the detection of visible (color) light and one for detecting NIR light emitted by the fluorescent probe. The detection of NIR, by definition due to the characteristics of the wavelength, is not related to the color of a drape or human tissue, but it is related to autofluorescence within tissue (chromophores) or cloths (natural fibers) which might be present in sterile drapes and is dependent on the fabric or detergent used. We tested several sterile drapes, of which the blue disposable ones were the less autofluorescent. In the future, multispectral imaging and therefore multispectral unmixing analytical techniques may correct for background autofluorescence also in the NIR region in order to improve sensitivity and specificity from a camera detection perspective.

Regarding the statement of the reviewer in which he states that: “the inability to use blue dye for sentinel node mapping is a departure from current standard of practice, in which dual tracer mapping is considered superior to single radionuclide mapping alone.” We do not agree with the statement of the reviewer. Recent literature has shown that only in <1% of the cases no radioactivity could be detected in the axilla when using isotopic staining, which means that only in rare cases additional use of a blue dye reduces the false negative rate⁶. In our study population, in all patients the lymphoscintigraphy showed clear uptake of the isotope in the axilla, and radioactivity

could be detected in all patients intraoperatively without the use of a blue dye. Moreover, in the Netherlands the clinical practice is shifting towards using only isotope staining for intraoperative lymph node mapping.

- *Additionally, do the limitations on ink color affect standard of care pathology practices for determining margin?*

Our analytical workflow is designed in such a way that it will fit in the standard of care with minimal changes. The limitation on ink color did not affect the standard of care pathology practices in both institutions participating in the study. Furthermore, new software algorithms are being designed that can correct for extrinsic 'fluorescence', which means that in the near future no restrictions on ink color will exist.

We have added a statement on page 21, lines 600-601.

- *Many surgeons utilize separate shave margins, a practice which already reduces re-operative rates for (+) margins and should be factored into the utility of this technology for achieving that goal.*

We agree with the reviewer that separate shave margins reduce positive margin rates, and these are already being used in the clinics mainly in the US. However, these cavity shavings are also known for overtreatment, because 70-80% of the patients do not have a positive surgical margin after lumpectomy. Which means that in 70-80% of the patients too much healthy tissue is unnecessarily removed. In a patient with a small breast volume a radical cavity shaving might result in a procedure close to a mastectomy, which cannot be regarded optimal health care if in 70-80% of the patients and can be considered overtreatment. Consequently, and by definition, in patients undergoing a lumpectomy it is very important to conserve as much healthy tissue to achieve the most optimal cosmetic result with a radical R0 tumor resection.

On the contrary to a total cavity shaving, fluorescence guided surgery can guide the surgeon to the areas where a selective cavity shaving needs to be performed, in other words: fluorescence guided surgery can be regarded a refinement tool for guided cavity shavings that might prevent overtreatment and not replacing the procedure as such by definition.

- *Near-tumor breast parenchymal tissue appears to demonstrate higher fluorescent intensity, which raises concerns for additional false positives.*

Although parenchymal tissue including collagen showed high tracer uptake at the *micro-segmentation analyses*, in the majority of patients the tumor signal is higher compared to parenchymal tissue signal intensities (see also answer on page 13 of this document). Furthermore, it is to be expected that parenchymal tissue including collagen will only attribute little relatively to background fluorescence intensity *intraoperatively* and will most likely not complicate *in vivo* surgical decision making

because this tissue type is mainly located in small streaks within large amounts of fat. Only large amounts of parenchymal tissue including collagen may influence the tumor-to-background ratio in vivo, which might be challenging in patients with a tumor directly behind the nipple and in premenopausal patients with dense breasts.

We have added this specific topic in the discussion section on page 13, line 358-369.

- *It is unclear who analyses the NIR imaging, whether it be the operating surgeon or an additional investigator present in theatre. Given the proposed clinical impact, there should be some discussion on operator training and inter-observer variability.*

We do agree with the reviewer that it is not extensively described how the intraoperative images were analyzed and by whom. All intraoperative images were retrospectively analyzed by one person: MK, medical doctor, and technical team member. She is fully trained to evaluate fluorescence images. We do agree with the reviewer that especially in next phase clinical trials where clinical endpoints are included, such as positive margin rate, it is very important to have surgeons trained to evaluate the fluorescence images to reduce the inter-observer variability. Therefore, we are currently developing a training program for surgeons and all study personnel participating in next phase clinical trials for fluorescent guided breast conserving surgery.

We clarified the methods of the analyses of the intraoperative images on page 23-24 line 676-688.

- *While it is mentioned in methods, there is no description of any adverse peri-operative events related to the use of bevacizumab (wound healing, etc.). Also, there is no mention of the additional patient/hospital burden of a pre-operative infusion/injection three days prior to the intended operation.*

We have included the (S)AE's in the patients characteristics table, Table 1. None of the patients felt any burden of the infusion three days prior to surgery and were highly motivated to participate in the study even when they had to travel for more than 2 hours to the hospital. Also, the burden for the hospital is negligible, since the infusion takes about 1,5 hours of time mainly because of the one-hour observation period. The latter is expected to be abandoned as in now more than 120 patients injected with bevacizumab-800CW, no (S)AEs have been observed related to the compound or imaging procedure.

This is added in the results section of the manuscript on page 5, line 101-105

- *Figure 5 repeats data from within the text, and Figure 6 outlines a strategy rather than original data.*

Figure 5 has changed into a standard 2x2 table (Table 2), as also suggested by reviewer #1.

We changed Figure 6 into a supplementary figure.

- *Table 1 lists an abbreviation (NST) that is not used within the table. Invasive ductal adenocarcinoma is listed as such in Table 1, but its alternative “invasive carcinoma of no specific type” is used in the text (Page 5, line 108).*

The right term is “invasive carcinoma of no specific type”, therefore we changed the term in Table 1 accordingly.

- *Table 1 also lists four patients in whom additional in situ components were identified on the margin. It is presumed that these are mutually exclusive from the four patients with a positive primary tumor margin, as there are eight total reported (+) throughout the text. It may be helpful to speak more in depth on the pathologic findings, i.e. were these contiguous to primary tumor or skip lesions.*

We do agree with the reviewer that it is important to speak in depth on the pathologic findings, therefore we already described this clearly in the original submitted manuscript: “In four patients, there was a tumor-involved surgical margin of the invasive primary tumor; in four other patients the surgical margin of unexpected additional carcinoma in situ was positive next to a completely removed primary tumor” (page 5 line)

- *All in all, a well done study, and worthy of further refinement and investigation, but perhaps prematurely ambitious in clinical impact.*

We do agree with the reviewer that the manuscript is prematurely ambitious in clinical impact, therefore we would like to change these statements in the proposal of reviewer #2 ‘indicate clinical feasibility in support of future studies to evaluate impact’.

References

1. Rosenthal EL, Warram JM, de Boer E, et al. Safety and Tumor Specificity of Cetuximab-IRDye800 for Surgical Navigation in Head and Neck Cancer. *Clinical Cancer Research*. 2015;21(16):3658-3666. doi:10.1158/1078-0432.CCR-14-3284.
2. Kuhlemann I, Kleemann M, Jauer P, Schweikard A, Ernst F. Towards X-ray free endovascular interventions - using HoloLens for on-line holographic visualisation. *Healthc Technol Lett*. 2017;4(5):184-187. doi:10.1049/htl.2017.0061.
3. Ellis LM, Hicklin DJ. VEGF-targeted therapy: mechanisms of anti-tumour activity. *Nat Rev Cancer*. 2008;8(8):579-591. doi:10.1038/nrc2403.

4. Linardou H, Kalogeras KT, Kronenwett R, et al. The prognostic and predictive value of mRNA expression of vascular endothelial growth factor family members in breast cancer: a study in primary tumors of high-risk early breast cancer patients participating in a randomized Hellenic Cooperative Oncology Group trial. *Breast Cancer Res.* 2012;14(6):R145. doi:10.1186/bcr3354.
5. Gaykema SBM, Brouwers AH, Lub-de Hooge MN, et al. 89Zr-bevacizumab PET imaging in primary breast cancer. *J Nucl Med.* 2013;54(7):1014-1018. doi:10.2967/jnumed.112.117218.
6. Rauch P, Merlin J-L, Leufflen L, et al. Limited effectiveness of patent blue dye in addition to isotope scanning for identification of sentinel lymph nodes: Cross-sectional real-life study in 1024 breast cancer patients. *International Journal of Surgery.* 2016;33(Part A):177-181. doi:10.1016/j.ijssu.2016.08.002.

REVIEWERS' COMMENTS:

Reviewer #1 (Remarks to the Author):

The authors have done an excellent job in responding in sufficient detail to the reviewers comments and questions and in making the suggested changes to the manuscript. This reviewer has no further questions.

Reviewer #2 (Remarks to the Author):

The reviewer thanks the authors for going to great length to address the comments. The manuscript now contains adequate technical details and results supporting the conclusions, and is much improved as a result.

Reviewer #3 (Remarks to the Author):

The authors have addressed all concerns. I believe the discussion of single vs dual tracer for sentinel lymph node biopsy may be rooted in the variations of technique around the world.